# Feature learning as alignment: a structural property of gradient descent in non-linear neural networks

**Daniel Beaglehole**[*,†]
*UC San Diego*
*dbeaglehole@ucsd.edu*

**Ioannis Mitliagkas**
*Mila, Université de Montréal*
*Google DeepMind*
*ioannism@google.com*

**Atish Agarwala**
*Google DeepMind*
*thetish@google.com*

**Reviewed on OpenReview:** *https://openreview.net/forum?id=JXCe2ZcUXr*

## Abstract

Understanding the mechanisms through which neural networks extract statistics from input-label pairs through feature learning is one of the most important unsolved problems in supervised learning. Prior works demonstrated that the gram matrices of the weights (the *neural feature matrices*, NFM) and the *average gradient outer products* (AGOP) become correlated during training, in a statement known as the neural feature ansatz (NFA). Through the NFA, the authors introduce mapping with the AGOP as a general mechanism for neural feature learning. However, these works do not provide a theoretical explanation for this correlation or its origins. In this work, we further clarify the nature of this correlation, and explain its emergence. We show that this correlation is equivalent to alignment between the left singular structure of the weight matrices and the newly defined *pre-activation tangent features* at each layer. We further establish that the alignment is driven by the interaction of weight changes induced by SGD with the pre-activation features, and analyze the resulting dynamics analytically at early times in terms of simple statistics of the inputs and labels. We prove the derivative alignment occurs almost surely in specific high dimensional settings. Finally, we introduce a simple optimization rule motivated by our analysis of the centered correlation which dramatically increases the NFA correlations at any given layer and improves the quality of features learned.

## 1 Introduction

Neural networks have emerged as the state-of-the-art machine learning methods for seemingly complex tasks, such as language generation (Brown et al., 2020), image classification (Krizhevsky et al., 2012), and visual rendering (Mildenhall et al., 2021). The precise reasons why neural networks generalize well have been the subject of intensive exploration, beginning with the observation that standard generalization bounds from statistical learning theory fall short of explaining their performance (Zhang et al., 2021).

A promising line of work emerged in the form of the neural tangent kernel, connecting neural networks to kernels in the wide limit (Jacot et al., 2018; Chizat et al., 2019). However, subsequent research showed that the success of neural networks relies critically on aspects of learning which are absent in kernel approximations (Ghorbani et al., 2019; Allen-Zhu and Li, 2019; Yehudai and Shamir, 2019; Li et al., 2020; Refinetti et al., 2021). Other

---

[*]Corresponding author.
[†]Work partially done as an intern at Google DeepMind.

work showed that low width suffices for gradient descent to achieve arbitrarily small test error (Ji and Zhu, 2020), further refuting the idea that extremely wide networks are necessary.

Subsequently, the success of neural networks has been largely attributed to *feature learning* - the ability of neural networks to learn representations of data which are useful for downstream tasks. However, the specific mechanism through which features are learned is an important unsolved problem in deep learning theory. A number of works have studied the abilities of neural networks to learn features in structured settings (Abbe et al., 2022; Ba et al., 2022; Nichani et al., 2023; Barak et al., 2022; Damian et al., 2022; Moniri et al., 2023; Parkinson et al., 2023). Some of that work proves strict separation in terms of sample complexity between neural networks trained with stochastic gradient descent and kernels (Mousavi-Hosseini et al., 2022).

The work above studies simple structure, such as learning from low-rank data or functions that are hierarchical compositions of simple elements. Recent work makes a big step towards generalizing these assumptions by proposing the *neural feature ansatz* (NFA) (Radhakrishnan et al., 2024a; Beaglehole et al., 2023), a general structure that emerges in the weights of trained neural networks. The NFA states that the gram matrix of the weights at a given layer (known as the *neural feature matrix* or NFM) is aligned with the *average gradient outer product* (AGOP) of the network with respect to the input to that layer. In particular, the NFM and AGOP are highly correlated in all layers of trained neural networks of general architectures, including practical models such as VGG (Simonyan and Zisserman, 2014), vision transformers (Dosovitskiy et al., 2021), and GPT-family models (Brown et al., 2020).

A major missing element is an explanation for *how and why* the AGOP and NFM become correlated through training with gradient descent. In this paper, we precisely describe the emergence of this correlation. We establish that the NFA is equivalent to alignment between the left singular structure of the weight matrices and the uncentered covariance of the *pre-activation tangent kernel* (PTK) (Section 2) features. We then introduce the *centered neural feature correlation* (C-NFC) which isolates this alignment process. We show empirically that the C-NFC is close to its maximum value of 1 at early times, and fully captures the NFA at late times for a variety of architectures (fully-connected, convolutional, and attention layers) over a diverse collection of datasets (Section 3). Our experiments suggest that the C-NFC drives the development of the NFA. Through this centering, we show that the dynamics of the C-NFC can be understood analytically in terms of the statistics of the data and labels at early times (Section 4). Using this decomposition, we show that the NFA emerges as a structural property of gradient descent (analytical result in the commonly studied setting of uniform data on the sphere, under certain assumptions on the activation and target functions). In particular, the first non-zero derivatives of the centered NFM and AGOP will be asymptotically identical. We further characterize how the NFA depends on the data distribution, and explore this analytically and experimentally.

Finally, motivated by our theory, we design an intervention to increase the influence of the C-NFC and make the NFA more robust: Speed Limited Optimization, a layerwise gradient normalization scheme (Section 5). The effectiveness of the latter update rule suggests a path towards rational design of architectures and training procedures that maximize the NFA notion of feature learning by promoting alignment dynamics.

## 2 Alignment between the weight matrices and the pre-activation tangents

In this section, we decompose the AGOP into the weight matrices and the feature covariance of the pre-activation tangent kernel (PTK), and demonstrate that the NFA is equivalent to alignment between the weights and these PTK features. We include a glossary of terms for ease of reference in Appendix A.

### 2.1 Preliminaries

We consider fully-connected neural networks with a single output of depth $L \geq 1$, where $L$ is the number of hidden layers, written $f : \mathbb{R}^d \to \mathbb{R}$. We write the input to layer $\ell \in \{0, \ldots, L\}$ as $x_\ell$, where $x_0 \equiv x$ is the original datapoint, and the pre-activation as $h_\ell(x)$. Then,

$$h_\ell(x) = W^{(\ell)} x_\ell, \qquad x_{\ell+1} = \phi(h_\ell(x)) , \qquad (1)$$

where $\phi$ is an element-wise nonlinearity, $W^{(\ell)} \in \mathbb{R}^{k_{\ell+1} \times k_\ell}$ is a weight matrix, and $k_\ell$ is the hidden dimension at layer $\ell$. We restrict $k_{L+1}$ to be the number of output logits, and set $k_0 = d$, where $d$ is the input dimension of the data. Note that $f(x) = h_L(x)$ and we ignore $x_{L+1}$. We train $f$ by gradient descent on a loss function $\mathcal{L}(\theta, X)$, where $X$ is an input dataset, and $\theta$ is the collection of weights.

We consider a supervised learning setup where we are provided $n$ input-label pairs $(x^{(1)}, y^{(1)}), \ldots, (x^{(n)}, y^{(n)}) \in \mathbb{R}^d \times \mathbb{R}$. We denote the network inputs (datapoints) $X \in \mathbb{R}^{n \times d}$ and the labels $y \in \mathbb{R}^{n \times 1}$. For a given network, the inputs to intermediate layers $\ell \in \{0, \ldots, L\}$ are written $X_\ell \in \mathbb{R}^{n \times k_\ell}$, where $X_0 \equiv X$. We train a fully-connected neural network to learn the mapping from network inputs to labels by minimizing a standard loss function, such as mean-squared-error or cross-entropy, on the dataset.

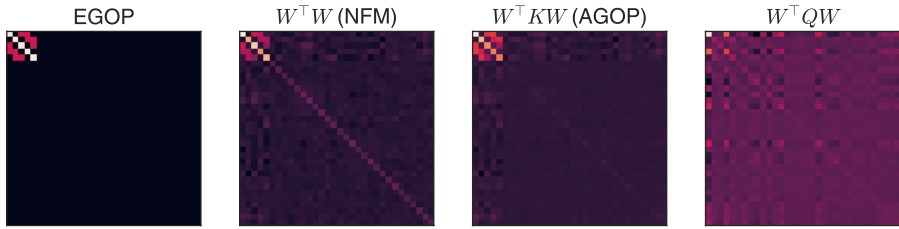

Figure 1: Various feature learning measures for target function $y(x) = \sum_{k=1}^{r} x_{k \bmod r} \cdot x_{(k+1) \bmod r}$ with $r = 5$ and inputs drawn from standard normal. The EGOP $\mathbb{E}_{x \sim \mu}\left[\frac{\partial y}{\partial x} \frac{\partial y}{\partial x}^\top\right]$ (first plot) captures the low-rank structure of the task. The NFM $\left(W^\top W\right)$ (second plot) and AGOP $\left(W^\top K W\right)$ (third plot) of a fully-connected network are similar to each other and the EGOP. Replacing $K$ with a symmetric matrix $Q$ with the same spectrum but independent eigenvectors obscures the low rank structure (fourth plot), and reduces the correlation from $\rho\left(F, \bar{G}\right) = 0.93$ to $\rho\left(F, W^\top Q W\right) = 0.53$.

One can define two objects associated with neural networks that capture learned structure. For a given layer $\ell$, the *neural feature matrix* (NFM) $F_\ell$ is the gram matrix of the columns of the weight matrix $W^{(\ell)}$, i.e. $F_\ell \equiv (W^{(\ell)})^\top W^{(\ell)}$. $F_\ell$ depends on the right singular vectors (and corresponding singular values) of $W^{(\ell)}$. The second fundamental object we consider is the *average gradient outer product* (AGOP) $\bar{G}_\ell$, defined as $\bar{G}_\ell \equiv \frac{1}{n} \sum_{\alpha=1}^{n} \frac{\partial f(x^{(\alpha)})}{\partial x_\ell} \frac{\partial f(x^{(\alpha)})}{\partial x_\ell}^\top$.

Since both these matrices have the same dimensions ($k_{\ell+1} \times k_{\ell+1}$), we can consider their cosine similarity. We define the *neural feature correlation* (NFC) by:

$$\rho\left(F_\ell, \bar{G}_\ell\right) \equiv \operatorname{tr}\left(F_\ell^\top \bar{G}_\ell\right) \cdot \operatorname{tr}\left(F_\ell^\top F_\ell\right)^{-1/2} \cdot \operatorname{tr}\left(\bar{G}_\ell^\top \bar{G}_\ell\right)^{-1/2} . \tag{NFC}$$

This takes on values in $[0, 1]$ since $F_\ell$ and $\bar{G}_\ell$ are both PSD.

Prior work has shown that in trained neural networks, the NFC will generally be close to 1 to varying degree of approximation. This notion is formalized in the *Neural Feature Ansatz* (NFA):

**Ansatz 1** (Neural Feature Ansatz (Radhakrishnan et al., 2024a))**.** *The Neural Feature Ansatz states that, for all layers $\ell \in [L]$ of a fully-connected neural network with $L$ hidden layers trained on input data $x^{(1)}, \ldots, x^{(n)}$, the NFC will satisfy $\rho\left(\bar{G}_\ell, F_\ell\right) \approx 1$ .*

The inputs to the covariance are the NFM and AGOP with respect to the input of layer $\ell$. Here, $\frac{\partial f(x)}{\partial x_\ell} \in \mathbb{R}^{k_\ell \times 1}$ denotes the gradient of the function $f$ with respect to the intermediate representation $x_\ell$. For simplicity, we may concatenate these gradients across the input dataset into a single matrix $\frac{\partial f(X)}{\partial x_\ell} \in \mathbb{R}^{n \times k_\ell}$. Note we consider scalar outputs in this work, though the NFA relation is identical when there are $c \geq 1$ outputs, where in this case $\frac{\partial f(x)}{\partial x_\ell} \in \mathbb{R}^{k_\ell \times c}$ is the input-output Jacobian of the model $f$.

We note that while the ansatz states exact proportionality, in practice, the NFM and AGOP are highly correlated with correlation less than 1, where the final correlation depends on many aspects of training and architecture choice (discussed further in Section 3).

The relation between the NFM and the AGOP is significant, in part, because for a neural network that has learned enough information about the target function, the AGOP of this model with respect to the first-layer inputs will approximate the *expected gradient outer product* (EGOP) of the target function (Yuan et al., 2023). In particular, the EGOP of the target function contains task-specific structure that is completely independent of the model used to estimate it. Where the labels are generated from a particular target function $y(x): \mathbb{R}^d \to \mathbb{R}$ on data sampled from a distribution $\mu$, the EGOP is defined as

$$\text{EGOP}(y, \mu) = \mathbb{E}_{x \sim \mu} \left[ \frac{\partial y}{\partial x} \frac{\partial y}{\partial x}^\top \right] .$$

If the neural feature ansatz (Ansatz 1) holds, the correlation of the EGOP and the AGOP at the end of training also implies high correlation between the NFM of the first layer and the EGOP, so that the NFM has encoded this task-specific structure.

To demonstrate the significance of high correlation between the NFM and the AGOP in successfully trained networks, we consider the following *chain-monomial* low-rank task:

$$y(x) = \sum_{k=1}^{r} x_{k \bmod r} \cdot x_{(k+1) \bmod r} , \tag{2}$$

where the data inputs are sampled from an isotropic Gaussian distribution $\mu = \mathcal{N}(0, I)$. In this case, the entries $\text{EGOP}(y, \mu)$ will be 0 for rows and columns outside of the $r \times r$ sub-matrix corresponding to $x_1, \ldots, x_r$ (Figure 1), as $y$ does not vary with coordinates $x_{r+1}, \ldots, x_d$. Within this sub-matrix, the diagonal entries will have value 2, while the off-diagonal entries will be either 1 or 0. Therefore, $\text{EGOP}(y, \mu)$ will be rank $r$, where $r$ is much less than the ambient dimension.

We verify for this task that the AGOP of the trained model resembles the EGOP (first and third panels of Figure 1). Here the NFA holds and the NFM (second panel) resembles the AGOP and therefore the EGOP as well. Therefore, the neural network has learned the model-independent and task-specific structure of the chain-monomial task in the right singular values and vectors of the first layer weight matrix, as these are determined by the NFM. In fact, previous works have demonstrated that the NFM of the first layer of a well-trained neural network is highly correlated with the AGOP of a fixed kernel method trained on the same dataset (Radhakrishnan et al., 2024a).

This insight has inspired iterative kernel methods which can match the performance of fully-connected networks (Radhakrishnan et al., 2024a;b; Aristoff et al., 2024) and improve over fixed convolutional kernels (Beaglehole et al., 2023). Additional prior works demonstrate the benefit of including the AGOP features to improve feature-less predictors (Hristache et al., 2001; Trivedi et al., 2014; Kpotufe et al., 2016). Additionally, because the NFM is correlated with the AGOP, the AGOP can be used to recover the features from feature-less methods, such as kernel machines. The AGOP has also been show to capture surprising phenomena of neural networks beyond low-rank feature learning including deep neural collapse (Beaglehole et al., 2024).

## 2.2 Alignment decomposition

In order to understand Ansatz 1, it is useful to decompose the AGOP. Doing so will allow us to show that the neural feature correlation (NFC) can be interpreted as an alignment metric between weight matrices and the *pre-activation tangent kernel* (PTK). The PTK $\mathcal{K}^{(\ell)}(x, z)$ is defined with respect to a layer $\ell$ of a neural network and two inputs $x, z$. The kernel evaluates to:

$$\mathcal{K}^{(\ell)}(x, z) \equiv \frac{\partial f(x)}{\partial h_\ell} \cdot \frac{\partial f(z)}{\partial h_\ell} . \tag{PTK}$$

When the arguments $x$ and $z$ are omitted, $\mathcal{K}^{(\ell)} \in \mathbb{R}^{n \times n}$ consists of the matrix of kernel evaluations on all pairs of datapoints. We may also consider the covariance $K^{(\ell)} \in \mathbb{R}^{k_\ell \times k_\ell}$ of the features associated with the PTK, $\frac{\partial f(X)}{\partial h_\ell} \in \mathbb{R}^{n \times k_\ell}$. In particular, we define the *feature covariance* of the PTK as,

$$K^{(\ell)} \equiv \frac{\partial f(X)}{\partial h_\ell}^\top \frac{\partial f(X)}{\partial h_\ell} . \tag{PTK feature covariance}$$

Crucially, for any layer $\ell$, we can re-write the AGOP in terms of this feature covariance as,

$$\bar{G}_\ell = (W^{(\ell)})^\top K^{(\ell)} W^{(\ell)} .$$

This gives us the following proposition:

**Proposition 2** (Alignment decomposition of NFC).

$$\rho\left(F_\ell, \bar{G}_\ell\right) = \rho\left((W^{(\ell)})^\top W^{(\ell)}, (W^{(\ell)})^\top K^{(\ell)} W^{(\ell)}\right) .$$

This alignment holds trivially and exactly if $K^{(\ell)}$ is the identity. However, the correlation can be high in trained networks where $K^{(\ell)}$ is non-trivial. For example, in the chain monomial task (Figure 1), $K^{(0)}$ is far from identity (standard deviation of its eigenvalues is 5.9 times its average eigenvalue), but the NFA correlation is 0.93 at the end of training. We also note that if $K^{(\ell)}$ is independent of $W^{(\ell)}$, the alignment is lower than in trained networks; in the same example, replacing $K^{(0)}$ with a matrix $Q$ with equal spectrum but random eigenvectors greatly reduces the correlation to 0.53 and qualitatively disrupts the structure relative to the NFM (Figure 1, rightmost column). We show the same result for the CelebA dataset (see Appendix M). Therefore, the NFA is a consequence of alignment between the left eigenvectors of $W^{(\ell)}$ and $K^{(\ell)}$ in addition to spectral considerations.

Note that $K^{(\ell)}$ itself is the AGOP of the neural network $f$ with respect to the pre-activations at layer $\ell$, i.e.

$$K^{(\ell)} \equiv \frac{1}{n} \sum_{\alpha=1}^{n} \frac{\partial f(x_\ell^{(\alpha)})}{\partial h_\ell} \frac{\partial f(x_\ell^{(\alpha)})}{\partial h_\ell}^\top . \tag{3}$$

For convolutional and attention layers, the $K^{(\ell)}$ are computed by additionally averaged over all patches and token positions in the input, respectively (see Radhakrishnan et al. (2024a); Beaglehole et al. (2023) for the formulation of the NFA in these architectures).

## 3 Centering the NFC isolates weight-PTK alignment

We showed that the neural feature ansatz is equivalent to PTK-weight alignment (Proposition 2). We now ask: is the increase in the NFC due to alignment of the weight matrices to the current PTK, or the alignment of the PTK to the current weights? In practice, both effects matter, but numerical evidence suggests that changes in the PTK do not drive the early dynamics of the NFC (Appendix D). Instead, we observe that the alignment between the weights and the PTK feature covariance is driven by a *centered* NFC that captures alignment between the *parameter changes* and the PTK. We then show this centered NFC can hold robustly in settings where the NFC holds with correlation less than 1, such as early in training and/or with large initialization. Finally, we establish analytically through the centered NFC how the NFA emerges as a structural property of gradient descent.

We begin by considering a decomposition of the NFM and AGOP into parts that depend on initialization and a part that depends on the changes in the weight matrix. Let $W_t^{(\ell)}$ and $K_t^{(\ell)}$ be the weight matrix and PTK feature covariance for layer $\ell$ at time $t$. We can write the NFM and AGOP in terms of the initial weights $W_0^{(\ell)}$ and the change in weights $\bar{W}_t^{(\ell)} \equiv W_t^{(\ell)} - W_0^{(\ell)}$ as follows:

$$\begin{aligned} F &= W_t^\top W_t = \bar{W}_t^\top \bar{W}_t + W_0^\top \bar{W}_t + \bar{W}_t^\top W_0 + W_0^\top W_0, \\ \bar{G} &= W_t^\top K_t W_t = \bar{W}_t^\top K_t \bar{W}_t + W_0^\top K_t \bar{W}_t + \bar{W}_t^\top K_t W_0 + W_0^\top K_t W_0 \end{aligned} \tag{4}$$

where we omitted $\ell$ from all terms for ease of notation. The first term in each decomposition isolates the gram matrix of the *changes* in the weight matrix. We call $\bar{W}^\top \bar{W}$ the *centered NFM* and $\bar{W}^\top K \bar{W}$ the *centered AGOP*. The centered AGOP in particular measures the alignment of weight updates with the current PTK feature covariance. Both terms are 0 at initialization, and if the weight matrices change significantly (that is, if $||\bar{W}|| \gg ||W_0||$), both the NFM and AGOP are dominated by the centered terms. (Note: in the limit that

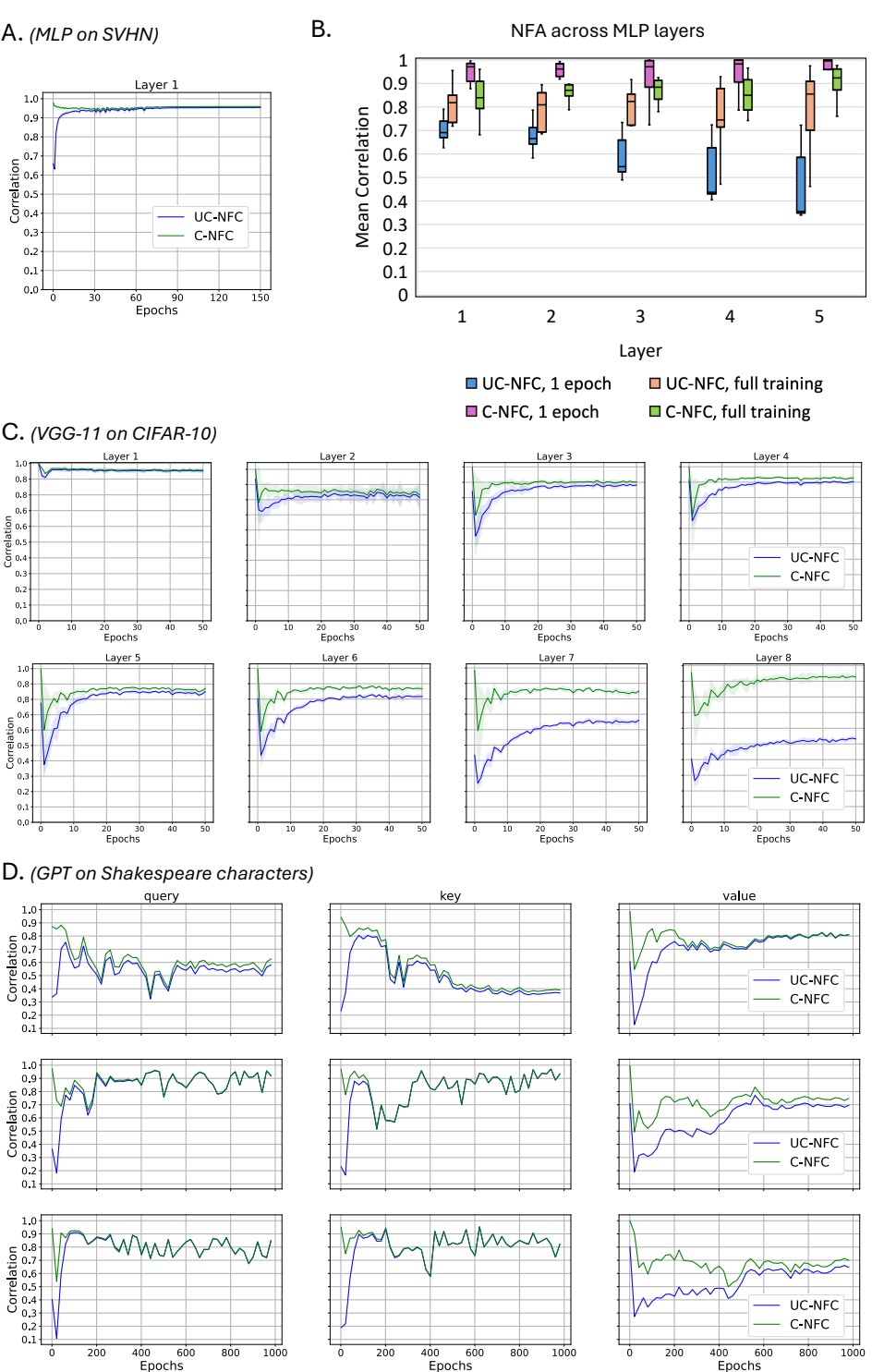

Figure 2: Uncentered and centered neural feature correlations across (A,B) fully-connected, (C) convolutional, and (D) attention layers with large initialization scale. (A,C,D) show trajectories of C/UC-NFC over training. (B) shows NFC values across all layers of an MLP with five hidden layers, averaged over CIFAR-10, CIFAR-100, SVHN, MNIST, GTSRB, and STL-10 datasets. (A-C) are additionally averaged over three random seeds. Each row of (D) is an attention block (ordered from first to last in the GPT model), while the columns show correlations for query, key, and value layers, respectively.

initialization scale vanishes to 0 (i.e. small initialization), then we expect the centered and uncentered NFM and AGOP to coincide, as $\bar{W} = W_t - W_0 \to W_t$, where $t$ is a step at the end of training.)

The decomposition in Equation 4 suggests that updates which drive correlation of the centered NFM and centered AGOP can also drive the overall value of the original NFC. Inspired by this observation, we define the *centered NFC* (C-NFC) as,

$$\rho\left((\bar{W}^{(\ell)})^\top \bar{W}^{(\ell)}, (\bar{W}^{(\ell)})^\top K^{(\ell)} \bar{W}^{(\ell)}\right) \ .$$

For the remainder of the paper, we will refer to the original NFC as the *uncentered NFC* (UC-NFC) to avoid ambiguity.

The centered NFC is consistently higher than the uncentered NFC across training times, architectures, and datasets (Figure 2). This especially holds at early times and in deeper layers of a network (Figure 2C and D, VGG-11 on CIFAR-10 and the GPT model on Shakespeare respectively). For MLPs, we conduct a broader set of experiments and verify that the trends hold on a wide range of vision datasets. We also note that the C-NFC at early times is relatively robust to the initialization statistics of the weight matrix $W^{(\ell)}$ - unlike the UC-NFC (for additional experiments, see Appendix H). We will return to this point in Section 5. These experiments suggest the C-NFC is responsible for improvements in the uncentered correlation.

High C-NFC values can drive increases in the UC-NFC as long as the centered NFM and centered AGOP are increasing in magnitude during training. We can confirm that the weights move significantly from initialization, which drives the contribution from the centered NFM and centered AGOP (Figures 14, and 15 in Appendix M). The increased importance of the C-NFC leads the UC-NFC to converge to the C-NFC at late times in most of our experimental settings, as the contribution from the weight changes dominates the contribution from the initialization. These findings validate that the C-NFC is an important contributor to increases in the UC-NFC.

Our experiments suggest that studying the C-NFC is a useful first step to understanding the neural feature ansatz. In Section 4, we develop a theoretical analysis to understand why the C-NFC is generically large at early times. We then use this analysis to predict the value of the C-NFC and motivate interventions which can keep the C-NFC high and promote the NFA earlier on in training (Section 5).

## 4 Theoretical analysis of the C-NFC at early times

### 4.1 Gradient flow dynamics

We now theoretically identify why the centered AGOP and NFM become correlated as a structural property of gradient descent, for at least the early training times. We consider the setup introduced in Equation (1). For theoretical convenience we focus on the case of training under gradient flow, where the dynamics of a weight matrix $W^{(\ell)}$ trained on loss $\mathcal{L}$ is given by

$$\dot{W}^{(\ell)} \equiv \frac{dW^{(\ell)}}{dt} = -\nabla_{W^{(\ell)}} \mathcal{L} \ , \tag{5}$$

where $\dot{W}^{(\ell)}$ is the time derivative of $W^{(\ell)}$. We expect our results to apply to gradient descent with small and intermediate learning rates as well. For simplicity of notation, we omit the layer index $\ell$.

We note that $\bar{W} = 0$. This gives us the following proposition, which shows that early time dynamics of the centered NFM and AGOP are dominated by the *second* time derivatives:

**Proposition 3** (Centered NFC dynamics). *Let $W$ be the weights of a fully-connected layer of a neural network at initialization and $X$ be the inputs to that layer. Then, when the neural network is trained by gradient flow on a loss function $\mathcal{L}$, we have $\bar{W}^\top \bar{W} = \bar{W}^\top K \bar{W} = 0$, $\frac{\mathrm{d}}{\mathrm{d}t}(\bar{W}^\top \bar{W}) = \frac{\mathrm{d}}{\mathrm{d}t}(\bar{W}^\top K \bar{W}) = 0$, and the first non-zero time derivatives satisfy,*

$$\frac{\mathrm{d}^2}{\mathrm{d}t^2}(\bar{W}^\top \bar{W}) = 2 \cdot X^\top \dot{\mathcal{L}} \mathcal{K} \dot{\mathcal{L}} X, \quad \frac{\mathrm{d}^2}{\mathrm{d}t^2}(\bar{W}^\top K \bar{W}) = 2 \cdot X^\top \dot{\mathcal{L}} \mathcal{K}^2 \dot{\mathcal{L}} X \tag{6}$$

*where $\dot{\mathcal{L}}$ is the $n \times n$ diagonal matrix of logit derivatives $\dot{\mathcal{L}} \equiv \mathrm{diag}\left(\frac{\partial \mathcal{L}}{\partial f}\right)$.*

*Proof of Proposition 3.* At initialization, all terms containing at least one copy of $\bar{W}$ vanish, leading to the zero and first time derivatives to be 0. Further, we have $\dot{W} = \frac{\partial f(X)}{\partial h}^\top \dot{\mathcal{L}} X$. Therefore, using that $K \equiv \frac{\partial f(X)}{\partial h}^\top \frac{\partial f(X)}{\partial h}$ and $\mathcal{K} \equiv \frac{\partial f(X)}{\partial h} \frac{\partial f(X)}{\partial h}^\top$,

$$\frac{1}{2}\frac{\mathrm{d}^2}{\mathrm{d}\,t^2}(\bar{W}^\top \bar{W}) = \dot{W}^\top \dot{W} = X^\top \dot{\mathcal{L}} \frac{\partial f(X)}{\partial h} \frac{\partial f(X)}{\partial h}^\top \dot{\mathcal{L}} X = X^\top \dot{\mathcal{L}} \mathcal{K} \dot{\mathcal{L}} X \ ,$$

and,

$$\begin{aligned}
\frac{1}{2}\frac{\mathrm{d}^2}{\mathrm{d}\,t^2}(\bar{W}^\top \bar{W}) = \dot{W}^\top K \dot{W} &= X^\top \dot{\mathcal{L}} \frac{\partial f(X)}{\partial h} K \frac{\partial f(X)}{\partial h}^\top \dot{\mathcal{L}} X \\
&= X^\top \dot{\mathcal{L}} \frac{\partial f(X)}{\partial h} \frac{\partial f(X)}{\partial h}^\top \frac{\partial f(X)}{\partial h} \frac{\partial f(X)}{\partial h}^\top \dot{\mathcal{L}} X \\
&= X^\top \dot{\mathcal{L}} \mathcal{K}^2 \dot{\mathcal{L}} X \ .
\end{aligned}$$

$\square$

We immediately see how gradient-based training can drive the C-NFC towards 1 at early times; the first non-trivial time derivatives are often highly correlated as they differ by only a single matrix power of $\mathcal{K}$, and have the same dependence on the labels. If $\mathcal{K}$ is proportional to a projection matrix, then the two derivatives will have perfect correlation; even if not, $\mathcal{K}$ and $\mathcal{K}^2$ will have identical eigenvectors, and hence we might expect a range of spectra will enable high correlation between them. In the case that $\mathcal{K}$ and $\mathcal{K}^2$ are sufficiently correlated, the C-NFC is driven to a high value immediately upon training, and the high value of the C-NFC eventually drives the UC-NFC and causes the NFA to hold.

In the remainder of this section, we study this correlation in two different high-dimensional limits. Our analysis suggests that for large models the C-NFC has a generic tendency to increase early in training.

## 4.2   Maximum C-NFC for data on the sphere

We first provide a general and well-studied setting where the first non-zero derivatives of the centered NFM and AGOP are perfectly correlated - uniform data on the sphere in high dimensions (Ghorbani et al., 2020; 2021; Misiakiewicz, 2022). This limit corresponds to an infinite width network which, combined with the data symmetry, induces the PTK matrix $\mathcal{K}$ to act like a projection matrix.

**Data setup**   We sample data $x \sim \sqrt{d} \cdot \mathrm{Unif}\left(\mathbb{S}^{d-1}\right)$ uniformly distributed on the sphere in $d$ dimensions with radius $\sqrt{d}$. We assume the labels are generated from a target function that maps $f^* : \mathbb{S}^{d-1} \to [-d, d]$. I.e. the label for the point $x$ is equal to $f^*(x)$. We train the parameters $a, W$ for a one-hidden layer fully-connected neural network $f(x) = f(x; a, W) = a^\top \phi(Wx)$ with element-wise activation function $\phi$. For a learning trajectory $(a_t, W_t)$, the initial values $a_0$ and $W_0$ are sampled i.i.d. standard Gaussian with variance at most $O(1/k)$. I.e. for all $i, j, \ell$, we have $a_0(i), W_0(j, \ell) \sim \mathcal{N}(0, c)$ for $c = O(1/k)$.

**Notation**   We will use asymptotic notation $O_d, o_d, \Theta_d, \Omega_d, \omega_d$ in the usual way, where the limits are taken with respect to the data dimension $d$. $\widetilde{O}_d, \widetilde{o}_d, \widetilde{\Theta}_d, \widetilde{\Omega}_d, \widetilde{\omega}_d$ are equivalent to their previous definitions but hide dependencies of polylogarithmic functions of $d$. We will use $\| \cdot \|$ to refer to $\ell_2$ vector norm and the operator norm of a matrix, i.e. $\|A\| = \max_{v \in \mathbb{R}^d} \frac{\|Av\|}{\|v\|}$, for a square matrix $A \in \mathbb{R}^{d \times d}$. Recall we sample $n$ datapoints and use hidden dimension $k$ so that $a_t \in \mathbb{R}^{k \times 1}$ and $W_t \in \mathbb{R}^{k \times d}$. We write the vector of all ones in $p$ dimensions as $\underline{1} \in \mathbb{R}^{p \times 1}$. We will omit the time index $t$ for ease of notation.

We make the assumption that $f^*$ has is bounded by a polylogarithmic factor of dimension with superlinearly vanishing probability:

**Assumption 4** (Labels bounded). *We have, for any positive $\delta > 0$, with probability $1 - O_d(d^{-1-\delta})$, $|f^*(x)| = O_d(\log^\ell(d))$ for a constant integer $\ell$.*

We also assume that the target function has a non-vanishing linear component:

**Assumption 5** (Non-trivial linear component). *We have, $\|\mathbb{E}_x[f^*(x)x]\| = \Omega_d(1)$.*

These assumptions hold for example when $f^*$ is exactly linear, i.e. $f^*(x) = \beta^\top x$ for $\|\beta\| \leq 1$. In this case, the inner product $x^\top \beta = \widetilde{O}_d(1)$ with probability $O_d(d^{-1-\delta})$ for $x$ uniform on $\sqrt{d} \cdot \mathbb{S}^{d-1}$ and any positive $\delta$.

We make an additional assumption on the differentiability of the PTK, inherited from the activation $\phi$, according to the conditions from Misiakiewicz (2022). This assumption is needed for our analysis, in which differentiability enables us to Taylor expand the PTK matrix (El Karoui, 2008). We restate this assumption.

**Assumption 6** (Differentiability of the PTK). *Let $\mathcal{K}(x, x') = \mathbb{E}_{a_k, w_k}[a_k^2 \phi'(w_k^\top x)\phi'(w_k^\top x')] = h_d(\langle x, x'\rangle/d)$ be the PTK for the network $f$, where $h_d : [-1, 1] \to \mathbb{R}$ is a positive semi-definite kernel function, defined for each dimension $d$. We assume there exist finite $h(0), h'(0), h''(0) > 0$ such that $\lim_{d\to\infty} h_d(0) = h(0)$, $\lim_{d\to\infty} h'_d(0) = h'(0)$, and $\lim_{d\to\infty} h''_d(0) = h''(0)$, where the first and second derivatives of $h_d$, $h'_d$ and $h''_d$, are assumed to exist on $[-1, 1]$ for all $d$.*

Note this is a sub-case of Assumption 1 in Misiakiewicz (2022) for level $\ell = 1$, and is satisfied for $h_d$ that is twice differentiable everywhere. We now introduce our final simplifying assumption on the activation function and data:

We will consider $f$ trained using mean-squared error (MSE) loss. With this loss, $\dot{\mathcal{L}}$ corresponds to the diagonal matrix of the residuals $y - f(x)$. If the outputs of the network is 0 on the training data, then $\dot{\mathcal{L}} = Y \equiv \mathrm{diag}(y)$, the labels themselves. We can guarantee this by either of the following methods:

**Method 7** (Subtract copy). *Subtracting off an identical (untrained) copy of the neural network from each output at initialization. I.e. we train the parameters of the network $f$ on the loss with respect to outputs $\hat{f}_t(x) = f_t(x) - f_0(x)$, where $f_0(x)$ is a copy of $f$ at initialization and $f_t(x) = f(x; a_t, W_t)$*

**Method 8** (Small initialization). *Initializing $W = 0$ or $\|a\| = \epsilon$ for $\epsilon > 0$ arbitrarily small.*

We now state our main theorem.

**Theorem 9** (Maximum C-NFC). *Suppose the data $X$ are sampled uniformly at random from $\mathbb{S}^{d-1}$ and the labels are generated by $f^*$ satisfying Assumptions 4 and 5. Suppose we train $f$ with MSE loss and initialize the network so that the initial outputs are 0 by either Method 7 or Method 8 above. Assume the activation function $\phi$ satisfies Assumption 6. We consider the regime that $\omega_d\left(d\log^{2\ell+1} d\right) \leq n \leq o_d\left(d^{2-\delta}\right)$ for some $\delta > 0$, and width $k \to \infty$. Then, at $t = 0$ in training (in the setting of Proposition 3), almost surely over the random $X$ as $n, d \to \infty$,*

$$\rho\left(\frac{\mathrm{d}^2}{\mathrm{d}\,t^2}(\bar{W}^\top \bar{W}), \frac{\mathrm{d}^2}{\mathrm{d}\,t^2}(\bar{W}^\top K \bar{W})\right) \to 1 \ .$$

The proof follows from the mean term of $\mathcal{K}$ giving the leading order terms in the C-NFC derivatives calculation here, and $\underline{1}\underline{1}^\top$ is a rank-1 projector. The proof is deferred to Appendix C.

We clarify that although we take infinite width, we are not necessarily in the NTK regime, as we allow for arbitrary scaling of the weights, including the $\mu$P parametrization (Yang and Hu, 2020). Hence, our setting allows for feature learning.

In the next section we will construct a dataset which interpolates between adversarial and aligned eigenstructure to demonstrate the range of possible values the derivative correlations can take, and theoretically predict their values.

### 4.3 Early time C-NFC dynamics in co-scaling regime

Another interesting limit is the linear co-scaling regime, where $n, d, k \to \infty$ with $n/d \equiv \psi_1$ and $k/d \equiv \psi_2$. Here, we show it is possible to theoretically predict the correlation of the derivatives of the centered NFM and AGOP.

In this regime, we expect the following two properties (Adlam and Pennington, 2020), which we state as assumptions. First, our key quantities can be written as traces of products of large random matrices; as the dimensions of all such matrices increase, we expect these traces to converge to their average values. This property is known as *self-averaging*.

**Assumption 10** (Self-averaging). *We assume that the expected traces appearing in the NFC across initializations are equal to the traces themselves.*

The elements of weight matrices $W$ are drawn from independent Gaussians (i.i.d. within each matrix). If the data $X$ were also standard Gaussian, we would be able to apply free probability — the noncommutative analog of classical independence — to compute traces of matrix products involving analytic functions of $W$ and $X$ in the limit of large dimensions (Mingo and Speicher, 2017). To apply free probability for more general $X$, we require the following assumption:

**Assumption 11** (Asymptotic freedom of initial parameters, and input-label pairs). *We assume $W$ and $X$ are asymptotically freely independent. Further, we assume the labels $Y(X)$ are asymptotically free of $W$ (but not $X$).*

For example, the labels in the student teacher setups of Adlam and Pennington (2020); Adlam et al. (2019) satisfy this condition.

From Proposition 3, we know that for MSE loss the correlation of NFM/AGOP time derivatives under gradient flow at initialization can be written as:

$$\rho\left(\dot{W}^\top \dot{W}, \dot{W}^\top K \dot{W}\right) = \operatorname{tr}\left(X^\top Y \mathcal{K} Y X X^\top Y \mathcal{K}^2 Y X\right) \cdot \operatorname{tr}\left((X^\top Y \mathcal{K} Y X)^2\right)^{-1/2} \cdot \operatorname{tr}\left((X^\top Y \mathcal{K}^2 Y X)^2\right)^{-1/2}. \quad (7)$$

In our high-dimensional limit, we expect that, for example, the first term can be given as

$$\lim_{n,d,k\to\infty} \operatorname{tr}\left(X^\top Y \mathcal{K} Y X X^\top Y \mathcal{K}^2 Y X\right) = \lim_{n,d,k\to\infty} \mathbb{E}_\theta\left[\operatorname{tr}\left(X^\top Y \mathcal{K} Y X X^\top Y \mathcal{K}^2 Y X\right)\right]. \quad (8)$$

We can decompose the average as

$$\mathbb{E}_\theta\left[\operatorname{tr}\left(X^\top Y \mathcal{K} Y X X^\top Y \mathcal{K}^2 Y X\right)\right] = \operatorname{tr}\left(X^\top Y \mathbb{E}_\theta\left[\mathcal{K}\right] Y X X^\top Y \mathbb{E}_\theta\left[\mathcal{K}^2\right] Y X\right) + \operatorname{tr}\left(\operatorname{Cov}\left(X^\top Y \mathcal{K} Y X, X^\top Y \mathcal{K}^2 Y X\right)\right) \quad (9)$$

with similar decompositions for the denominator term.

Therefore if the statistics of $\mathcal{K}$ can be understood as a function of $X$, we can compute the correlation in this linear triple-scaling limit. We focus for now on a one-hidden layer quadratic network (similar to the previous section) to avoid the branching of terms that arises in more complicated networks. We provide some additional analysis of the first term in Equation 9 for more complicated architectures in Appendix E.

### 4.4 Exact predictions with one hidden layer and quadratic activations

Concretely, we study a neural network $f$ which can be written as $f(x) = a^\top (Wx)^2$, where $a$ is the readout layer, and the square is element-wise. Let $M_{X|Y}^{(4)} = (X^\top Y X)^2$, and $M_X^{(2)} = X^\top X$, and $F_a = W^\top \operatorname{diag}\left(a^2\right) W$. Then, the numerator of Equation 9 can be written as:

$$\operatorname{tr}\left(X^\top Y \mathcal{K} Y X X^\top Y \mathcal{K}^2 Y X\right) = \operatorname{tr}\left(M_{X|Y}^{(4)} F_a M_{X|Y}^{(4)} F_a M_X^{(2)} F_a\right).$$

We reiterate our assumptions and compute this trace (as well as those for the denominator terms) in Appendix F using standard results from random matrix theory. These calculations show us that the correlation is determined by traces of powers of $M_{X|Y}^{(4)}$ and $M_X^{(2)}$ by the calculations in Section F, which are properties of the input-label pairs, and $F_a$, which is specific to the architecture and initialization statistics.

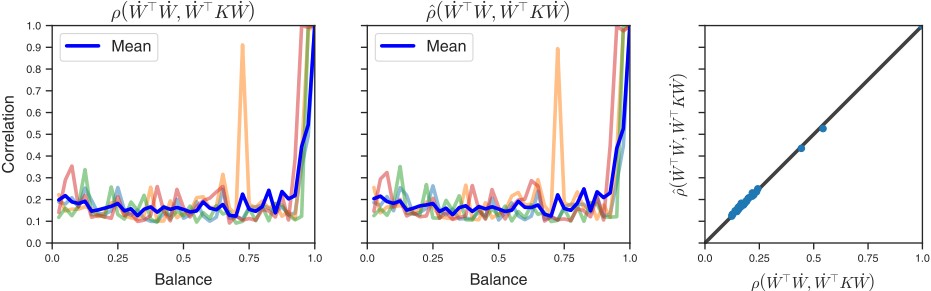

Figure 3: Predicted versus observed correlation of the second derivatives of centered $F$ and $\bar{G}$ on the alignment reversing dataset. Different shaded color curves correspond to four different seeds for the dataset. The solid blue curve is the average over all data seeds. The rightmost sub-figure is a scatter plot of the predicted versus observed correlations of these second derivatives, with one point for each balance value. We instantiate the dataset in the proportional regime where width, input dimension, and dataset size are all equal to 1024.

**Manipulating the C-NFC**  To numerically explore the validity of the random matrix theory calculations, we developed a method to generate datasets with different values of $\rho\left(\dot{W}^{\top}\dot{W}, \dot{W}^{\top}K\dot{W}\right)$. We construct a random dataset called the *alignment reversing* dataset, parameterized by a *balance* parameter $\gamma \in (0, 1]$ to adversarially disrupt the NFA near initialization in the regime that width $k$, input dimension $d$, and dataset size $n$ are all equal ($n = k = d = 1024$). By Proposition 16, for the aforementioned neural architecture, the expected second derivative of the centered NFM satisfies, $\mathbb{E}\left[\dot{W}^{\top}\dot{W}\right] = X^{\top}Y\mathbb{E}\left[\mathcal{K}\right]YX = (X^{\top}YX)^2$, while the expected second derivative of the centered AGOP, $\mathbb{E}\left[\dot{W}^{\top}K\dot{W}\right] = X^{\top}Y\mathbb{E}\left[\mathcal{K}^2\right]YX$, has an additional component $X^{\top}YX \cdot X^{\top}X \cdot X^{\top}YX$. Our construction exploits this difference in that $X^{\top}X$ becomes adversarially unaligned to $X^{\top}YX$ as the balance parameter decreases.

The construction exploits that we can manipulate $X^{\top}YX \cdot X^{\top}X \cdot X^{\top}YX$ freely of the NFM using a certain choice of $Y$. We design the dataset such that this AGOP-unique term is close to identity, while the NFM second derivative has many large off-diagonal entries, leading to low correlation between the second derivatives of the NFM and AGOP.

In our experiment, we sample multiple random datasets with this construction and compute the predicted and observed correlation of the second derivatives of the centered NFC at initialization. For specific details on the construction, see Appendix G.

We observe in Figure 3 that the centered NFC predicted with random matrix theory closely matches the observed values, across individual four random seeds and for the average of the correlation across them. Crucially, a single neural network is used across the datasets, confirming the validity of the self-averaging assumption. The variation in the plot across seeds come from randomness in the sample of the data, which cause deviations from the adversarial construction.

## 5   Increasing the centered contribution to the NFC strengthens feature learning

Our theoretical and experimental work has established that gradient based training leads to alignment of the weight matrices to the PTK feature covariance. This process is driven by the C-NFC. Therefore, one path towards improving the neural feature correlations is to increase the contribution of the C-NFC to the dynamics - as measured, for example, by the centered-to-uncentered, or, C/UC, ratio $\operatorname{tr}\left(\bar{W}^{\top}\bar{W}\bar{W}^{\top}K\bar{W}\right)\operatorname{tr}\left(W^{\top}WW^{\top}KW\right)^{-1}$. When this ratio is large, the C-NFC contributes significantly in magnitude to the UC-NFC, indicating successful feature learning as measured by the uncentered NFC. We will discuss how small initialization promotes feature learning, and design an optimization rule, Speed Limited Optimization, that increases the C/UC ratio and drives the value of the UC-NFC to 1.

### 5.1 Feature learning and initialization

One factor that modulates the level of feature learning is the scale of the initialization in each layer. In our experiments training networks with unmodified gradient descent, we observe that the centered NFC will increasingly dominate the uncentered NFC with decreasing initialization (third panel, first row, Figure 4). Further, as the centered NFC increases in contribution to the uncentered quantity, the strength of the UC-NFC, and to a lesser extent, the C-NFC increases (Figure 4). The decreases in correspondence between these quantities is also associated with a decrease in the feature quality of the NFM (Appendix I), for the chain monomial task.

For fully-connected networks with homogeneous activation functions, such as ReLU, and no normalization layers, decreasing initialization scale is equivalent to decreasing the scale of the outputs, since we can write $f(Wx) = a^{-p}f(aWx)$ for any scalar $a$ for any homogenous activation $f$. This in turn is equivalent to increasing the scale of the labels. Therefore, decreasing initialization forces the weights to change more in order to fit the labels, leading to more change in $F$ from its initialization. Conceptually, this aligns with the substantial line of empirical and theoretical evidence that increasing initialization scale or output scaling transitions training between the lazy and feature learning regimes (Chizat et al., 2019; Woodworth et al., 2020; Agarwala et al., 2020; Lyu et al., 2023).

This relationship suggests that small initialization can be broadly applied to increase the change in F, and the value of the UC-NFC. However, this may not be ideal; for example, if the activation function is differentiable at 0, small initialization leads to a network which is approximately linear. This may lead to low expressivity unless the learning dynamics can increase the weight magnitude.

### 5.2 Speed Limited Optimization

We can instead design an intervention which can increase feature learning without the need to decrease the initialization scale. We do so by fixing the learning *speed* layerwise to constant values, which causes the C-NFC to dominate the UC-NFC dynamics. For weights at layer $\ell$ and learning rate $\eta > 0$, we introduce *Speed Limited Optimization (SLO)*, which is characterized by the following update rule,

$$W_{t+1}^{(\ell)} \leftarrow W_t^{(\ell)} - \eta \cdot C_\ell \cdot \frac{\nabla_{W_t^{(\ell)}}\mathcal{L}}{\|\nabla_{W_t^{(\ell)}}\mathcal{L}\|}$$

where the hyperparameter $C_\ell \geq 0$ controls the amount of learning in layer $\ell$. We expect this rule to increase the strength of the UC-NFC in layers where $C_\ell$ is large relative to $C_m$ for $m \neq \ell$, as $W^{(\ell)}$ will be forced to change significantly from initialization. By forcing the weights in a particular layer $\ell$ to have fixed learning speeds, these weights will have fixed updates sizes at every epoch, regardless of the loss. We downscale the speeds in other layers $\ell \neq m$ to prevent training instability. As a result, $\left\|W_0^{(\ell)}\right\| \left\|W_t^{(\ell)}\right\|^{-1} \to 0$ for large $t$, causing the centered and uncentered NFC to coincide for this layer.

We demonstrate the effects of SLO on the chain-monomial task (Equation (2)). We found that fixing the learning speed to be high in the first layer and low in the remaining layers causes the ratio of the unnormalized C-NFC to the UC-NFC to become close to 1 across initialization scales (Figure 4). The same result holds for the SVHN dataset (see Appendix M). We note that this intervention can be applied to target underperforming layers and improve generalization in deeper networks (see Appendix K for details).

We observe that both the C-NFC and the UC-NFC become close to 1 after training with SLO, independent of initialization scale (Appendix I). Further, the quality of the features learned, measured by the similarity of $F$ and $\bar{G}$ to the true EGOP, significantly improve and are more similar to each other with SLO, even with large initialization. In contrast, in standard training the UC-NFA fails to develop with large initialization, as $F$ resembles identity (no feature learning), while $\bar{G}$ only slightly captures the relevant features.

Speed Limited Optimization is a step toward the design of optimizers that improve generalization and training times through maximizing feature learning.

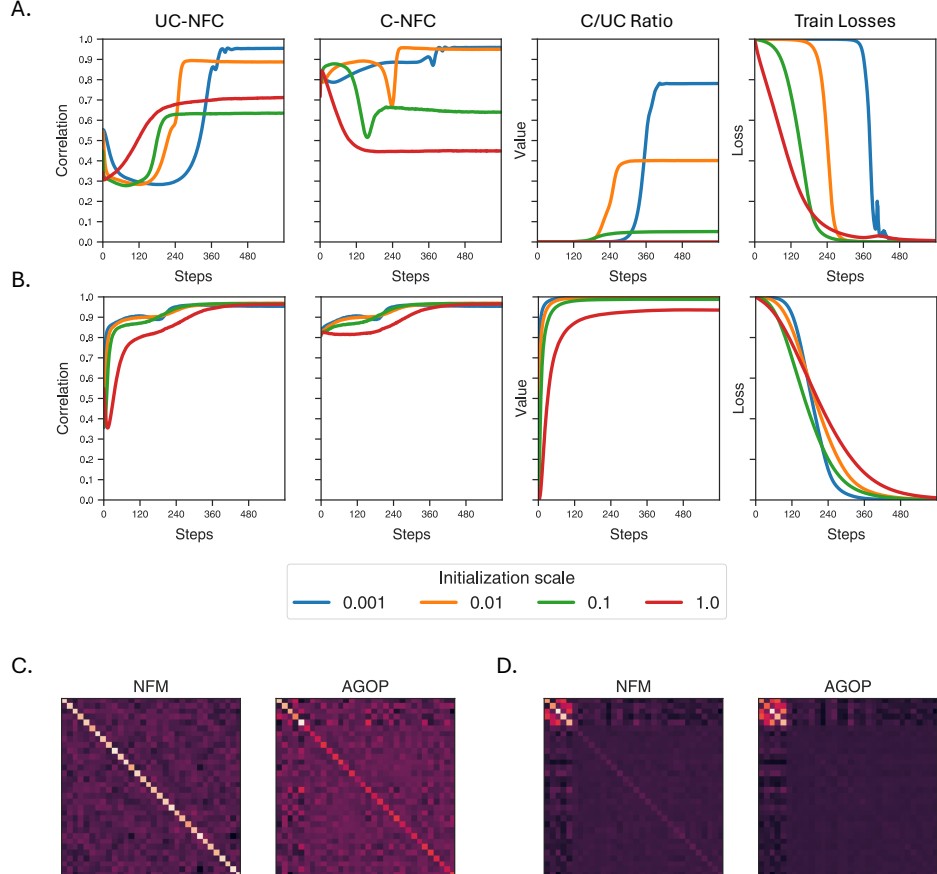

Figure 4: The effect of SLO on C/UC neural feature correlations and feature learning on the chain monomial task. In the first two rows, we plot the uncentered and centered NFA for the first layer weight matrix as a function of initialization, (A) with standard training and (B) with SLO. We consider a two hidden layer network with ReLU activations, where we set $C_0 = 500$, and $C_1 = C_2 = 0.002$. The third column shows the ratio of the unnormalized C-NFC to the UC-NFC: $\text{tr}\left(\bar{W}^\top \bar{W} \bar{W}^\top K \bar{W}\right) \cdot \text{tr}\left(W^\top W W^\top K W\right)^{-1}$. The fourth column shows the training loss. In the third row, we plot the NFM and AGOP from a trained network with (C) standard training and (D) with Speed Limited Optimization with fixed initialization scale of 1.0.

# 6 Discussion

**Analyzing more general settings**   In principle our analysis can be extended to a larger sample size and activation functions with uncentered derivatives. Both of these settings require analyzing more terms in the Taylor expansion of the PTK matrix. This is more complicated than studying the loss, as our most complex calculations require understanding degree 8 polynomials of the inputs even in the simplest case of a one-hidden layer network (as opposed to degree 4 to understand the loss). We may also want to understand the case that $n = d^\ell$ for integer $\ell$ (i.e. without the logarithmic factor we consider). The appropriate Taylor expansions in these cases are discussed in Misiakiewicz (2022), and will likely require additional structure on the coefficients of the target function $f^*$ in the basis of spherical harmonics.

**Centered NFC through training**   Our theoretical analysis in this work shows that the PTK feature covariance at initialization has a relatively simple structure in terms of the weights and the data. To predict the NFC later in training, we will likely need to account for the change in this matrix. One should be able to predict this development at short times by taking advantage of the fact that eigenvectors of the Hessian change slowly during training (Bao et al., 2023). An alternative approach would be to use a quadratic model

for neural network dynamics (Agarwala et al., 2022; Zhu et al., 2022) which can capture dynamics of the empirical NTK (and therefore the PTK). One promising avenue to study the time dynamics of the PTK could be to adapt the results of Wang et al. (2024), which study how the effects of feature learning can propagate to the conjugate kernel for neural networks. It may also be useful to unify our notion of alignment with the observations of Arous et al. (2023), where they find that the weights align with the averaged outer products of the loss gradients.

**Speed Limited Optimization**    There are prior works that implement differential learning rates across layers (Howard and Ruder, 2018; Singh et al., 2015), though these differ from differential learning *speeds* as in SLO, in which the L2-norm of the gradients are fixed, and not the scaling of the gradients. Our intervention is more similar to the LARS (You et al., 2017) and LAMB (You et al., 2019) optimizers that fix learning speeds to decrease training time in ResNet and BERT architectures. However, these optimizers fix learning speeds to the norm of the weight matrices, while our intervention sets learning speeds to free hyperparameters in order to increase feature learning, as measured by the NFC. We also point out that as an additional explanation for the success of SLO in our experiments, the PTK feature covariance may change slowly (or just in a single direction) with this optimizer, because the learning rate is small for later layers, allowing the first layer weights to align with the PTK.

Other works consider interventions that vary the strength of feature learning across layers. The authors of Yang and Hu (2020) down-scale the final layer weights so that the weights in earlier layers must move significantly in $\ell_\infty$ norm to fit the data. The SLO intervention is more extreme than this approach in that SLO overrides the natural GD dynamics to exactly set the weight movement rates. In Chizat and Netrapalli (2024) the authors consider adjusting learning rates in each layer so as to force the activation vectors to substantially change across all layers. Instead, SLO sets learning speeds of the weights in each layer so that some weights move much faster than others, depending on the layer where we want to maximize feature learning.

**Adaptive and stochastic optimizers**    We demonstrate in this work that the dataset (by our construction in Section 4) and optimizer (with SLO in Section 5) play a significant role in the strength of feature learning. Important future work would be to understand other settings where significant empirical differences exist between optimization choices. In particular, analyzing the NFC may clarify the role of gradient batch size and adaptive gradient methods in generalization (Zhu et al., 2023).

## 7    Acknowledgements

We thank Lechao Xiao for detailed feedback on the manuscript. We also thank Jeffrey Pennington for helpful discussions. This work used the programs (1) XSEDE (Extreme science and engineering discovery environment) which is supported by NSF grant numbers ACI-1548562, and (2) ACCESS (Advanced cyberinfrastructure coordination ecosystem: services & support) which is supported by NSF grants numbers #2138259, #2138286, #2138307, #2137603, and #2138296. Specifically, we used the resources from SDSC Expanse GPU compute nodes, and NCSA Delta system, via allocations TG-CIS220009.

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

## A Glossary

1. **Expected gradient outer product (EGOP).** Defined with respect to a target function $f^*$ and an input data distribution $\mu$ over data in $d$ dimensions.

$$\text{EGOP}(f^*, \mu) \equiv \mathbb{E}_{x \sim \mu} \left[ \nabla_x f^*(x) \nabla_x f^*(x)^\top \right] \in \mathbb{R}^{d \times d}.$$

2. **Average gradient outer product (AGOP).** Defined with respect to a predictor $f$ and a set of inputs $\{x^{(1)}, \ldots, x^{(n)}\}$.

$$\text{AGOP}(f, \{x^{(\alpha)}\}_\alpha) \equiv \frac{1}{n} \sum_{\alpha=1}^{n} \frac{\partial f(x^{(\alpha)})}{\partial x} \frac{\partial f(x^{(\alpha)})}{\partial x}^\top \in \mathbb{R}^{d \times d}.$$

In the context of a deep neural network, we write the AGOP $\bar{G}_\ell$ with respect to the inputs of a given layer $\ell$ as

$$\bar{G}_\ell \equiv \frac{1}{n} \sum_{\alpha=1}^{n} \frac{\partial f(x^{(\alpha)})}{\partial h_\ell} \frac{\partial f(x^{(\alpha)})}{\partial h_\ell}^\top.$$

3. **Neural feature matrix (NFM).** Given a neural network $f$ with a weight matrix $W^{(\ell)}$ at layer $\ell$, the NFM $F_\ell$ is defined as,

$$F_\ell \equiv (W^{(\ell)})^\top W^{(\ell)}.$$

4. **Neural feature correlation (NFC).** Defined with respect to a layer of a neural network, this is the correlation $\rho\left(F_\ell, \bar{G}_\ell\right)$.

5. **Neural feature ansatz (NFA).** This refers to the statement that the NFC will have correlation approximately equal to 1.

6. **Pre-activation tangent kernel (PTK).** This is the kernel function corresponding, defined with respect to a neural network and a layer $\ell$, that evaluates two inputs $x, z$ to $\frac{\partial f(x)}{\partial h_\ell} \cdot \frac{\partial f(z)}{\partial h_\ell} \in \mathbb{R}$.

7. **PTK feature covariance.** For a given layer of a neural network and a dataset $X$, the PTK feature covariance $K^{(\ell)}$ is defined as $K^{(\ell)} \equiv \frac{\partial f(X)}{\partial h_\ell}^\top \frac{\partial f(X)}{\partial h_\ell} \in \mathbb{R}^{k_\ell \times k_\ell}$.

8. **Centered NFC (C-NFC).** Consider a layer $W$ of a neural network with PTK feature covariance $K$ with respect to that layer. Let $\bar{W} \equiv W - W_0$, where $W_0$ are weights at initialization. Then, the C-NFC is equivalent to the correlation $\rho\left(\bar{W}^\top \bar{W}, \bar{W}^\top K \bar{W}\right)$.

9. **Uncentered NFC (UC-NFC).** This quantity is identical to the NFC defined above.

# B Connection of the PTK to the empirical NTK

We note the following connection between the PTK entries and the empirical NTK (ENTK). In particular, the PTK is a significant component of the ENTK.

**Proposition 12** (Pre-activation to neural tangent identity)**.** *Consider a depth-$L$ neural network $f(x)$ with inputs $X_\ell \in \mathbb{R}^{n \times k_\ell}$ to weight matrices $W^{(\ell)} \in \mathbb{R}^{k_\ell \times k_\ell}$ for $\ell \in [L]$. Consider the empirical NTK where gradients are taken only with respect to $W^{(\ell)}$ evaluated between two points $x$ and $z$ denoted by $\hat{\Theta}_\ell(x_0, z_0) = \langle \frac{\partial f(x_0)}{\partial W^{(\ell)}}, \frac{\partial f(z_0)}{\partial W^{(\ell)}} \rangle$. Then for all $\ell$, $\hat{\Theta}_\ell$ satisfies,*

$$\hat{\Theta}_\ell(x_0, z_0) = \mathcal{K}^{(\ell)}(x_0, z_0) \cdot x_\ell^\top z_\ell \ .$$

*Proof of Proposition 12.* Note that $\frac{\partial f(x_0)}{\partial W^{(\ell)}} = \frac{\partial f(x_0)}{\partial h_\ell} x_0^\top \in \mathbb{R}^{k_\ell \times k_\ell}$. Then,

$$\begin{aligned}
\hat{\Theta}_\ell(x_0, z_0) &= \operatorname{tr}\left( \frac{\partial f(x_0)}{\partial h_\ell} x_\ell^\top z_\ell \frac{\partial f(z_0)}{\partial h_\ell}^\top \right) \\
&= \frac{\partial f(x_0)}{\partial h_\ell}^\top \frac{\partial f(z_0)}{\partial h_\ell} \cdot x_\ell^\top z_\ell \\
&= \mathcal{K}^{(\ell)}(x_0, z_0) \cdot x_\ell^\top z_\ell \ .
\end{aligned}$$

$\square$

# C Additional proofs and statements

**Theorem** (Maximum C-NFC)**.** *Suppose the data $X$ are sampled uniformly at random from $\mathbb{S}^{d-1}$ and the labels are generated by $f^*$ satisfying Assumptions 4 and 5. Suppose we train $f$ with MSE loss and initialize the network so that the initial outputs are $0$ by either Method 7 or Method 8 above. Assume the activation function $\phi$ satisfies Assumption 6. We consider the regime that $\omega_d\left(d \log^{2\ell+1} d\right) \le n \le o_d\left(d^{2-\delta}\right)$ for some $\delta > 0$, and width $k \to \infty$. Then, at $t = 0$ in training (in the setting of Proposition 3), almost surely over the random $X$ as $n, d \to \infty$,*

$$\rho\left( \frac{\mathrm{d}^2}{\mathrm{d} t^2}(\bar{W}^\top \bar{W}), \frac{\mathrm{d}^2}{\mathrm{d} t^2}(\bar{W}^\top K \bar{W}) \right) \to 1 \ .$$

*Proof of Theorem 9.* Applying Proposition 3 and using that we train with MSE and zero initial outputs, we have

$$\frac{1}{2} \frac{\mathrm{d}^2}{\mathrm{d} t^2}(\bar{W}^\top \bar{W}) = X^\top Y \mathcal{K} Y X, \quad \frac{1}{2} \frac{\mathrm{d}^2}{\mathrm{d} t^2}(\bar{W}^\top \bar{W}) = X^\top Y \mathcal{K}^2 Y X \ .$$

Given that $k \to \infty$, $\mathcal{K}$ is equal to a deterministic kernel matrix conditioned on the inputs $X$ (i.e. does not depend on the sampled initial weights). We then use that we are in the proportional limit to approximate the kernel matrix $\mathcal{K}$ by its Taylor expansion (Misiakiewicz, 2022; El Karoui, 2008; Hu and Lu, 2022; Ba et al., 2019). Namely, we have that with probability $1 - o_d(1)$,

$$\mathcal{K} = \gamma \underline{1} \underline{1}^\top + \frac{\alpha}{d} X X^\top + \mu I + o_d(1) \cdot \Delta \ , \tag{10}$$

where $\gamma = h_d(0) + \frac{1}{d} h_d''(0)$, $\alpha = h'(0)$, $\mu = h(1) - h(0) - h'(0)$, and $\Delta$ with $\|\Delta\|_2 = 1$ is a matrix. In general, we will write standalone asymptotic variables to indicate matrices $\Delta$ of that order spectral norm.

By the linearization in equation (10) and the structure of $X X^\top$, we will show,

$$\mathcal{K} = \Theta(1) \cdot \underline{1} \underline{1}^\top + \widetilde{O}(1) \cdot \Delta_1, \quad \mathcal{K}^2 = \Theta(d) \cdot \underline{1} \underline{1}^\top + \widetilde{O}(d) \cdot \Delta_2 \ ,$$

where $\Delta_1, \Delta_2$ have spectral norm 1. To prove this, first note that $\frac{1}{n}X^\top X = I + o(1)$ as uniform data on the sphere are $O(1)$-subgaussian and have identity covariance. Therefore, as $n = \widetilde{\Theta}(d)$, and the spectra of $X^\top X$ and $XX^\top$ are identical, we have that $\|\frac{1}{d}XX^\top\| = \|\frac{1}{d}X^\top X\| = \widetilde{O}(1)$.

We now analyze the numerator in the correlation (using the normalized trace). To simplify notation we write $\langle A, B \rangle \equiv \operatorname{tr}\left(A^\top B\right)$ for matrices $A, B$. The numerator is then:

$$\langle X^\top Y\mathcal{K}YX, X^\top Y\mathcal{K}^2 YX \rangle = \widetilde{\Theta}(d) \cdot \operatorname{tr}\left((X^\top yy^\top X)^2\right)$$
$$+ \widetilde{O}(d) \cdot \operatorname{tr}\left((X^\top Y\Delta_1 YX)(X^\top yy^\top X)\right)$$
$$+ \widetilde{O}(d) \cdot \operatorname{tr}\left((X^\top yy^\top X)(X^\top Y\Delta_2 YX)\right)$$
$$+ \widetilde{O}(d) \cdot \operatorname{tr}\left((X^\top Y\Delta_1 YX)(X^\top Y\Delta_2 YX)\right)$$

As the labels are bounded, $\|Y\Delta_1 Y\|, \|Y\Delta_2 Y\| = \widetilde{O}(1)$. Therefore, by the sub-multiplicative property of the spectral norm and that $\|X^\top\|\|X\| = \|X^\top X\|$, we have $\|X^\top Y\Delta_1 YX\| \leq \|Y\Delta_1 Y\|\|X^\top X\| = \widetilde{O}(1) \cdot \|X^\top X\|$ and $\|X^\top Y\Delta_2 YX\| = \widetilde{O}(1) \cdot \|X^\top X\|$. Finally, again applying $\|X^\top X\| = \widetilde{O}(d)$,

$$\operatorname{tr}\left((X^\top Y\Delta_1 YX)(X^\top yy^\top X)\right) \leq \widetilde{O}(1) \cdot \|X^\top X\| \cdot \|X^\top y\|^2 = \widetilde{O}(d) \cdot \|X^\top y\|^2,$$

and,

$$\operatorname{tr}\left((X^\top yy^\top X)(X^\top Y\Delta_2 YX)\right) \leq \widetilde{O}(d) \cdot \|X^\top y\|^2.$$

The fourth trace then satisfies,

$$\operatorname{tr}\left((X^\top Y\Delta_1 YX)(X^\top Y\Delta_2 YX)\right) \leq \widetilde{O}(1) \cdot \|X^\top X\|^2 = \widetilde{O}(d^2)$$

Combining the four traces and using that $\operatorname{tr}\left((X^\top yy^\top X)^2\right) = \operatorname{tr}\left((y^\top XX^\top y)^2\right) = \|X^\top y\|^4$,

$$\langle X^\top Y\mathcal{K}YX, X^\top Y\mathcal{K}^2 YX \rangle = \widetilde{\Omega}(d) \cdot \|X^\top y\|^4 + \widetilde{O}(d^2) \cdot \|X^\top y\|^2 + \widetilde{O}(d^3) .$$

Recall by Assumption 5, we have $\|\mathbb{E}_x\left[xf^*(x)\right]\| = \Omega(1)$. We apply Lemma 13 for our choice of $n$ so that $\|\frac{1}{n}X^\top y - \mathbb{E}\left[xf^*(x)\right]\| = o(1)$ w.p. $1 - o(1)$. As a consequence, we have $\|X^\top y\| = \widetilde{\Omega}(d)$ w.h.p. Therefore, the leading term in the numerator of the correlation is that of $\operatorname{tr}\left((X^\top yy^\top X)^2\right)$ - the term due to the interaction of the mean terms of $\mathcal{K}$ and $\mathcal{K}^2$.

Meanwhile the first denominator term decomposes as follows:

$$\langle X^\top Y\mathcal{K}YX, X^\top Y\mathcal{K}YX \rangle = \widetilde{\Theta}(1) \cdot \operatorname{tr}\left((X^\top yy^\top X)^2\right)$$
$$+ \widetilde{O}(1) \cdot \operatorname{tr}\left((X^\top Y\Delta_1 YX)(X^\top yy^\top X)\right)$$
$$+ \widetilde{O}(1) \cdot \operatorname{tr}\left((X^\top Y\Delta_1 YX)^2\right)$$

By a similar argument to the numerator term, we observe that the leading sub-term for the first denominator term is $\widetilde{\Theta}(1) \cdot \operatorname{tr}\left((X^\top yy^\top X)^2\right)$.

Further, the second denominator terms decomposes as follows:

$$\langle X^\top Y\mathcal{K}^2 YX, X^\top Y\mathcal{K}^2 YX \rangle = \widetilde{\Theta}(d^2) \cdot \operatorname{tr}\left((X^\top yy^\top X)^2\right)$$
$$+ \widetilde{O}(d^2) \cdot \operatorname{tr}\left((X^\top Y\Delta_2 YX)(X^\top yy^\top X)\right)$$
$$+ \widetilde{O}(d^2) \cdot \operatorname{tr}\left((X^\top Y\Delta_2 YX)^2\right)$$

By a similar argument, the second denominator term has leading sub-term $\Theta(d^2) \cdot \operatorname{tr}\left((X^\top yy^\top X)^2\right)$.

Putting the numerator and denominator terms together, we have the correlation has the following asymptotics:

$$\rho\left(\frac{\mathrm{d}^2}{\mathrm{d}\,t^2}(\bar{W}^\top\bar{W}), \frac{\mathrm{d}^2}{\mathrm{d}\,t^2}(\bar{W}^\top K\bar{W})\right) = \frac{\widetilde{\Theta}(d)\cdot\mathrm{tr}\left((X^\top yy^\top X)^2\right) + \widetilde{O}(d^3)}{\mathrm{tr}\left(\widetilde{\Theta}(1)\cdot(X^\top yy^\top X)^2 + \widetilde{O}(d^2)\right)^{1/2}\cdot\mathrm{tr}\left(\widetilde{\Theta}(d^2)\cdot(X^\top yy^\top X)^2 + \widetilde{O}(d^4)\right)^{1/2}}$$

$$\to 1\;,$$

completing the proof. □

**Lemma 13.** *Suppose $|f^*(x)| \le c\cdot\log^\ell(d)$ with probability $1 - o(1)$ for constant $c > 0$, and the data are sub-Gaussian with constant parameter $K$ and some $\ell > 0$. Then, $\|\frac{1}{n}X^\top y - \mathbb{E}\left[xf^*(x)\right]\| = o(1)$ w.p. $1 - o(1)$, provided $n = \omega(d\log^{2\ell+1}(d))$.*

*Proof.* As the data $x$ are $K$-sub-Gaussian for a universal constant $K$, and the labels are bounded by $c\log^\ell(d)$, we have that $xy$ is sub-Gaussian with parameter $M = Kc\log^\ell(d)$. Therefore, the empirical expectation of $xy$, $\frac{1}{n}\sum_{i=1}^n x_i y_i$ is sub-Gaussian with parameter $M/\sqrt{n}$. Directly applying Lemma 1 in Jin et al. (2019), we see that,

$$\Pr\left(\left\|\frac{1}{n}\sum_{i=1}^n x_i y_i - \mathbb{E}\left[xy\right]\right\| > t\right) < 2\exp\left(-\frac{t^2 n}{2cM^2 d}\right)\;,$$

for a universal constant $c > 0$. Therefore, provided $n = \omega(d\log^{2\ell+1}(d))$, we have $\left\|\frac{1}{n}\sum_{i=1}^n x_i y_i - \mathbb{E}\left[xy\right]\right\| = o(1)$ with probability $1 - o(1)$, completing the proof. □

## D   Additional centerings of the NFC

**Double-centered NFC**  One may additionally center the PTK feature map to understand the co-evolution of the PTK feature covariance and the weight matrices. We consider such a centering that we refer to as the *double-centered* NFC, where we measure $\rho\left((\bar{W}^{(\ell)})^\top\bar{W}^{(\ell)}, (\bar{W}^{(\ell)})^\top\bar{K}^{(\ell)}\bar{W}^{(\ell)}\right)$, where $\bar{K}^{(\ell)} = \left(\frac{\partial f(X)}{\partial h_\ell} - \frac{\partial f_0(X)}{\partial h_\ell}\right)^\top\left(\frac{\partial f(X)}{\partial h_\ell} - \frac{\partial f_0(X)}{\partial h_\ell}\right)$, and $f_0$ is the neural network at initialization.

However, the double-centered NFC term corresponds to higher-order dynamics that do not significantly contribute the centered and uncentered NFC (Figure 5) when initialization is large or for early periods of training. Note however this term becomes relevant over longer periods of training.

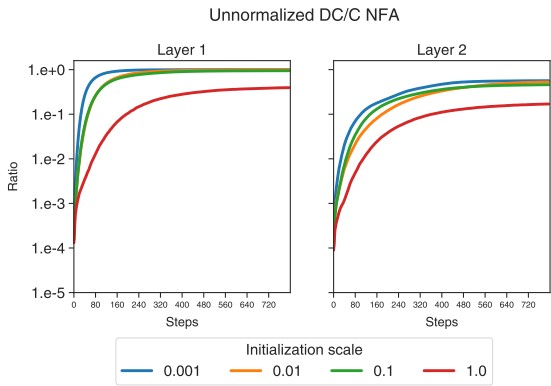

Figure 5: Ratio of the unnormalized double-centered NFC to the centered NFC throughout neural network training. In particular, we plot $\mathrm{tr}\left(\bar{W}^\top\bar{W}\bar{W}^\top\bar{K}\bar{W}\right)\cdot\mathrm{tr}\left(\bar{W}^\top\bar{W}\bar{W}^\top K\bar{W}\right)^{-1}$ throughout training for both layers of a two-hidden layer MLP with ReLU activations.

**Isolating alignment of the PTK to the initial weight matrix** One may also center just the PTK feature map, while substituting the initial weights for $W$ to isolate how the PTK feature covariance aligns to the weight matrices. To measure this alignment, we consider the *PTK-centered* NFC, which is defined as the correlation $\rho\left((W_0^{(\ell)})^\top W_0^{(\ell)}, (W_0^{(\ell)})^\top \bar{K}^{(\ell)} W_0^{(\ell)}\right)$, where $W_0^{(\ell)}$ is the initial weight matrix at layer $\ell$.

However, this correlation decreases through training, indicating that the correlation of these quantities does not drive alignment between the uncentered NFM and AGOP (Figure 6).

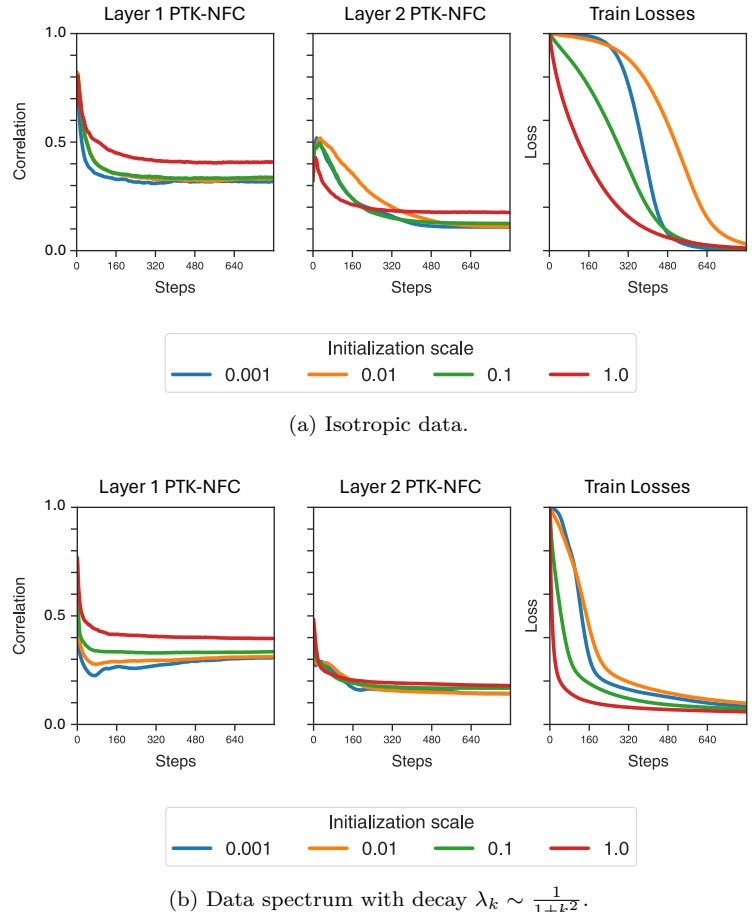

(a) Isotropic data.

(b) Data spectrum with decay $\lambda_k \sim \frac{1}{1+k^2}$.

Figure 6: PTK-centered NFC throughout training for both layers of a two-hidden layer MLP with ReLU activations on Gaussian data with two different spectra.

# E  Extending our theoretical predictions to depth and general activations

Precise predictions of the C-NFC become more complicated with additional depth and general activation functions. However, we note that the deep C-NFC will remain sensitive to a first-order approximation in which $K$ is replaced by its expectation. We demonstrate that this term qualitatively captures the behavior of the C-NFC for 2 hidden layer architectures with quadratic and, to a lesser extent, ReLU activation functions in Figure 7. In this experiment, we sample Gaussian data with mean 0 and covariance with a random eigenbasis. We parameterize the eigenvalue decay of the covariance matrix by a parameter $\alpha$, called the data decay rate, so that the eigenvalues have values $\lambda_k = \frac{1}{1+k^\alpha}$. As $\alpha$ approaches 0 or $\infty$ the data covariance approaches a projector matrix.

In this experiment, we see that the data covariance spectrum will also parameterize the eigenvalue decay of $\mathbb{E}\left[K\right]$, allowing us to vary how close the expected PTK matrix (and its dual, the PTK feature covariance) is

from a projector, where the NFA holds exactly. We see that for intermediate values of $\alpha$, both the observed and the predicted derivatives of the C-NFC decreases in value.

We plot the observed values in two settings corresponding to different asymptotic regimes. One setting is the proportional regime, where $n = k = d = 128$. The other is the NTK regime where $n = d = 128$ and $k = 1024$. For the quadratic case, as the network approaches infinite width, the prediction more closely matches the observed values. Additional terms corresponding to the nonlinear part of $\phi'$ in ReLU networks, the derivative of the activation function, are required to capture the correlation more accurately in this case.

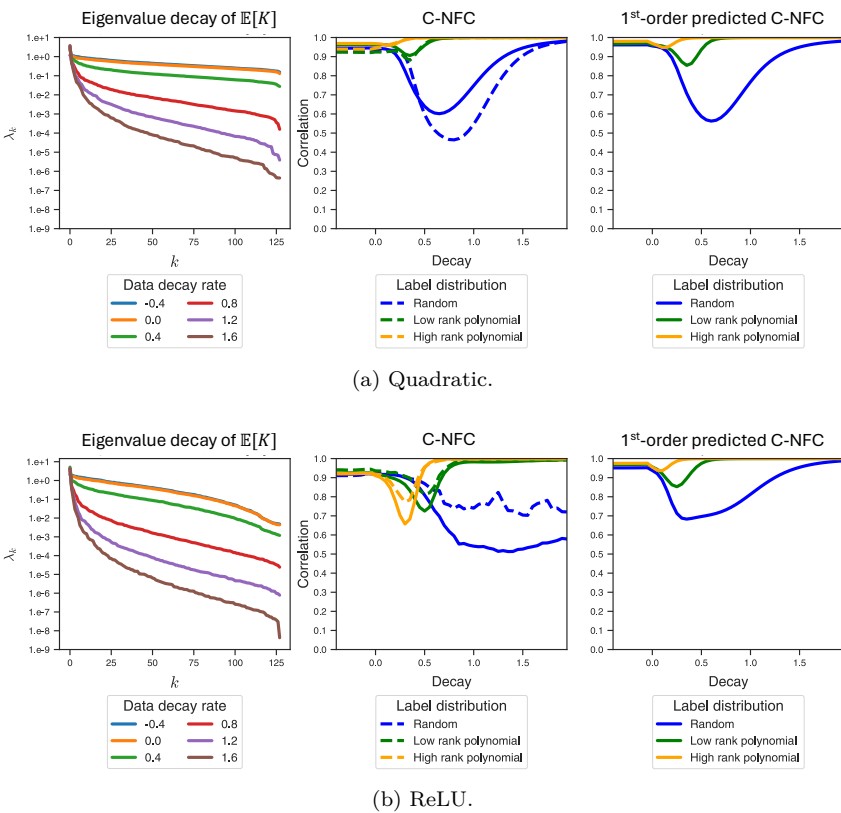

Figure 7: Observed versus the first-order predicted C-NFC for the input to the first layer of a two hidden layer MLP. The dashed line is neural network width $k = n = d = 128$, where $n$ and $d$ are the number of data point and data dimension, respectively, while the solid line uses $n = d = 128$ and $k = 1024$.

# F   Free probability calculations of C-NFC

In order to understand the development of the NFC, we analyze the centered NFC in the limit that learning rate is much smaller than the initialization for a one hidden layer MLP with quadratic activations. We write this particular network as,

$$f(x) = a^\top (Wx)^2 \ ,$$

where $a \in \mathbb{R}^{1 \times k}$ and $W \in \mathbb{R}^{k \times d}$, where $d$ is the input dimension and $k$ is the width. In this case, the NFC has the following form,

$$\rho\left(F, \bar{G}\right) = \frac{\operatorname{tr}\left(X^\top Y \mathcal{K} Y X X^\top Y \mathcal{K}^2 Y X\right)}{\operatorname{tr}\left((X^\top Y \mathcal{K} Y X)^2\right)^{-1/2} \operatorname{tr}\left((X^\top Y \mathcal{K}^2 Y X)^2\right)^{-1/2}} \ , \tag{11}$$

where $\mathcal{K} = X W^\top \operatorname{diag}\left(a\right)^2 W X^\top$.

We assume two properties hold in the finite dimensional case we consider, that will hold asymptotically in the infinite dimensional limit.

**Assumption 14** (Self-averaging). *We assume that computing the average of the NFC quantities across initializations is equal to the quantities themselves in the high-dimensional limit.*

**Assumption 15** (Asymptotic freeness). *We assume that the collections $\{X, Y\}$ and $\{W, a\}$ are asymptotically free with respect to the operator $\mathbb{E}\left[\mathrm{tr}(\cdot)\right]$, where $\mathrm{tr}[M] = \frac{1}{n}\sum_{i=1}^{n} M_{ii}$.*

We will compute the expected values of the centered NFC under these assumptions. In the remainder of the section we will drop the $\mathbb{E}\left[\cdot\right]$ in the trace for ease of notation.

### F.1 Free probability identities

The following lemmas will be useful: let $\{\bar{C}_i\}$ and $\{R_i\}$ be freely independent of each other with respect to tr, with $\mathrm{tr}[\bar{C}_i] = 0$. Alternating words have the following products:

$$\mathrm{tr}[\bar{C}_1 R_1] = 0 \tag{12}$$

$$\mathrm{tr}[\bar{C}_1 R_1 \bar{C}_2 R_2] = \mathrm{tr}[R_1]\mathrm{tr}[R_2]\mathrm{tr}[\bar{C}_1 \bar{C}_2] \tag{13}$$

$$\mathrm{tr}[\bar{C}_1 R_1 \bar{C}_2 R_2 \bar{C}_3 R_3] = \mathrm{tr}[R_1]\mathrm{tr}[R_2]\mathrm{tr}[R_3]\mathrm{tr}[\bar{C}_1 \bar{C}_2 \bar{C}_3] \tag{14}$$

$$\begin{aligned}
\mathrm{tr}[\bar{C}_1 R_1 \bar{C}_2 R_2 \bar{C}_3 R_3 \bar{C}_4 R_4] = {}& \mathrm{tr}[R_1]\mathrm{tr}[R_2]\mathrm{tr}[R_3]\mathrm{tr}[R_4]\mathrm{tr}[\bar{C}_1 \bar{C}_2 \bar{C}_3 \bar{C}_4] + \\
& \mathrm{tr}[R_1]\mathrm{tr}[R_3]\mathrm{tr}[\bar{R}_2 \bar{R}_4]\mathrm{tr}[\bar{C}_1 \bar{C}_2]\mathrm{tr}[\bar{C}_3 \bar{C}_4] + \mathrm{tr}[R_2]\mathrm{tr}[R_4]\mathrm{tr}[\bar{R}_1 \bar{R}_3]\mathrm{tr}[\bar{C}_2 \bar{C}_3]\mathrm{tr}[\bar{C}_1 \bar{C}_4]
\end{aligned} \tag{15}$$

where $\bar{R}_i \equiv R_i - \mathrm{tr}[R_i]$.

Applying these identities to the one hidden layer quadratic case, we use the following definitions:

$$R = W^\top \mathrm{diag}\left(a^2\right) W, \ \ A = (X^\top Y X)^2, \ \ B = X^\top X \tag{16}$$

Crucially, $R$ is freely independent of the set $\{A, B\}$. We will also use the notation $\bar{M}$ to indicated the centered version of $M$, $\bar{M} = M - \mathrm{tr}[M]$.

### F.2 Numerator term of NFC

The numerator in Equation (11) is

$$\mathrm{tr}\left(X^\top Y \mathcal{K} Y X X^\top Y \mathcal{K}^2 Y X\right) = \mathrm{tr}\left(ARARBR\right) \tag{17}$$

Re-writing $A = \bar{A} + \mathrm{tr}[A]$ and $B = \bar{B} + \mathrm{tr}[B]$ we have:

$$\mathrm{tr}\left(X^\top Y \mathcal{K} Y X X^\top Y \mathcal{K}^2 Y X\right) = \mathrm{tr}\left((\bar{A} + \mathrm{tr}[A])R(\bar{A} + \mathrm{tr}[A])R(\bar{B} + \mathrm{tr}[B])R\right) \tag{18}$$

This expands to

$$\begin{aligned}
\mathrm{tr}\left(X^\top Y \mathcal{K} Y X X^\top Y \mathcal{K}^2 Y X\right) = {}& \mathrm{tr}\left(\bar{A}R\bar{A}R\bar{B}R\right) + 2\mathrm{tr}\left(A\right)\mathrm{tr}\left(\bar{A}R\bar{B}R^2\right) + \mathrm{tr}\left(B\right)\mathrm{tr}\left(\bar{A}R\bar{A}R^2\right) \\
& \mathrm{tr}\left(A\right)^2\mathrm{tr}\left(\bar{B}R^3\right) + 2\mathrm{tr}\left(A\right)\mathrm{tr}\left(B\right)\mathrm{tr}\left(\bar{A}R^3\right) + \mathrm{tr}\left(A\right)^2\mathrm{tr}\left(B\right)\mathrm{tr}\left(R\right)^3
\end{aligned} \tag{19}$$

Using the identities we arrive at:

$$\begin{aligned}
\mathrm{tr}\left(X^\top Y \mathcal{K} Y X X^\top Y \mathcal{K}^2 Y X\right) = {}& \mathrm{tr}[A]^2\mathrm{tr}[B]\mathrm{tr}\left(R^3\right) \\
& + 2\mathrm{tr}[A]\mathrm{tr}[R^2]\mathrm{tr}[R]\mathrm{tr}\left(\bar{A}\bar{B}\right) \\
& + \mathrm{tr}[B]\mathrm{tr}[R]\mathrm{tr}[R^2]\mathrm{tr}\left(\bar{A}^2\right) \\
& + \mathrm{tr}[R]^3\mathrm{tr}\left(\bar{A}^2\bar{B}\right)
\end{aligned} \tag{20}$$

### F.3 First denominator term of NFC

The first denominator term in Equation (11) is

$$\mathrm{tr}\left(X^\top Y \mathcal{K} Y X X^\top Y \mathcal{K} Y X\right) = \mathrm{tr}\left(ARAR\right) \tag{21}$$

This is a classic free probability product:

$$\mathrm{tr}\left(X^\top Y \mathcal{K} Y X X^\top Y \mathcal{K} Y X\right) = \mathrm{tr}[A^2]\mathrm{tr}[R]^2 + \mathrm{tr}[A]^2\mathrm{tr}\left(R^2\right) - \mathrm{tr}[A]^2\mathrm{tr}[R]^2 \tag{22}$$

which can be derived from the lemmas.

### F.4 Second denominator term of NFC

For the second denominator term of Equation (11) we have

$$\mathrm{tr}\left(X^\top Y \mathcal{K}^2 Y X X^\top Y \mathcal{K}^2 Y X\right) = \mathrm{tr}\left(ARBRARBR\right) \tag{23}$$

Expanding the first $A$ we have

$$\mathrm{tr}\left(X^\top Y \mathcal{K}^2 Y X X^\top Y \mathcal{K}^2 Y X\right) = \mathrm{tr}\left(\bar{A}RBRARBR\right) + \mathrm{tr}[A]\mathrm{tr}\left(R^2 BRARB\right) \tag{24}$$

Next we expand the first $B$:

$$\begin{aligned}
\mathrm{tr}\left(X^\top Y \mathcal{K}^2 Y X X^\top Y \mathcal{K}^2 Y X\right) &= \mathrm{tr}\left(\bar{A}R\bar{B}RARBR\right) + \mathrm{tr}[B]\mathrm{tr}\left(\bar{A}R^2 ARBR\right) \\
&\quad + \mathrm{tr}[A]\mathrm{tr}[B]\mathrm{tr}\left(R^3 ARB\right) + \mathrm{tr}[A]\mathrm{tr}\left(R^2 \bar{B}RARB\right)
\end{aligned} \tag{25}$$

The next $A$ gives us

$$\begin{aligned}
\mathrm{tr}\left(X^\top Y \mathcal{K}^2 Y X X^\top Y \mathcal{K}^2 Y X\right) &= \mathrm{tr}\left(\bar{A}R\bar{B}R\bar{A}RBR\right) + 2\mathrm{tr}[A]\mathrm{tr}\left(\bar{A}R\bar{B}R^2 BR\right) + \mathrm{tr}[B]\mathrm{tr}\left(\bar{A}R^2\bar{A}RBR\right) \\
&\quad + 2\mathrm{tr}[A]\mathrm{tr}[B]\mathrm{tr}\left(R^3\bar{A}RB\right) + \mathrm{tr}[A]^2\mathrm{tr}[B]\mathrm{tr}\left(R^4 B\right) + \mathrm{tr}[A]^2\mathrm{tr}\left(R^2\bar{B}R^2 B\right)
\end{aligned} \tag{26}$$

Expanding the final $B$ we have

$$\begin{aligned}
\mathrm{tr}\left(X^\top Y \mathcal{K}^2 Y X X^\top Y \mathcal{K}^2 Y X\right) &= \mathrm{tr}\left(\bar{A}R\bar{B}R\bar{A}R\bar{B}R\right) + 2\mathrm{tr}[B]\mathrm{tr}\left(\bar{A}R\bar{B}R\bar{A}R^2\right) + 2\mathrm{tr}[A]\mathrm{tr}\left(\bar{A}R\bar{B}R^2\bar{B}R\right) \\
&\quad + 4\mathrm{tr}[A]\mathrm{tr}[B]\mathrm{tr}\left(R^3\bar{A}R\bar{B}\right) + 2\mathrm{tr}[A]\mathrm{tr}[B]^2\mathrm{tr}\left(R^4\bar{A}\right) + 2\mathrm{tr}[A]^2\mathrm{tr}[B]\mathrm{tr}\left(R^4\bar{B}\right) \\
&\quad + \mathrm{tr}[A]^2\mathrm{tr}[B]^2\mathrm{tr}[R^4] + \mathrm{tr}[A]^2\mathrm{tr}[R^2\bar{B}R^2\bar{B}] + \mathrm{tr}[B]^2\mathrm{tr}[R^2\bar{A}R^2\bar{A}]
\end{aligned} \tag{27}$$

Now all terms are in the form of alternating products from the lemma. This means we can factor out the non-zero traces of the other terms. Simplifying we have:

$$\begin{aligned}
\mathrm{tr}\left(X^\top Y \mathcal{K}^2 Y X X^\top Y \mathcal{K}^2 Y X\right) &= \mathrm{tr}[R]^4\mathrm{tr}\left((\bar{A}\bar{B})^2\right) + 2\mathrm{tr}[R]^2(\mathrm{tr}[R^2] - \mathrm{tr}[R]^2)\mathrm{tr}[\bar{A}\bar{B}]^2 \\
&\quad + 2\mathrm{tr}[R]^2\mathrm{tr}[R^2]\left(\mathrm{tr}[B]\mathrm{tr}\left(\bar{A}^2\bar{B}\right) + \mathrm{tr}[A]\mathrm{tr}\left(\bar{A}\bar{B}^2\right)\right) \\
&\quad + 4\mathrm{tr}[A]\mathrm{tr}[B]\mathrm{tr}[R^3]\mathrm{tr}[R]\mathrm{tr}\left(\bar{A}\bar{B}\right) + \mathrm{tr}[A]^2\mathrm{tr}[B]^2\mathrm{tr}[R^4] \\
&\quad + \mathrm{tr}[A]^2\mathrm{tr}[\bar{B}^2]\mathrm{tr}[R^2]^2 + \mathrm{tr}[\bar{A}]^2\mathrm{tr}[B^2]\mathrm{tr}[R^2]^2
\end{aligned} \tag{28}$$

All terms of the NFC are now in terms of traces of the matrices $A$, $B$, and $R$ and functions on each term separately. The matrices $A$ and $B$ are determined by the data, while the moments of the eigenvalues of $R$ are determined by the initialization distribution of the weights in the neural network, and neither training nor the data.

# G   Alignment reversing dataset

The data consists of a mixture of two distributions from which two subsets of the data $X^{(1)}$ and $X^{(2)}$ are sampled from, and is parametrized by a balance parameter $\gamma \in (0, 1]$ and two variance parameters $\epsilon_1, \epsilon_2 > 0$. The subset $X^{(1)}$ which has label $y_1 = 1$ and constitutes a $\gamma$ fraction of the entire dataset, is sampled from a multivariate Gaussian distribution with mean 0 and covariance $\Sigma = \underline{1}\,\underline{1}^\top + \epsilon_1 \cdot I$ . Then the second subset, $X^{(2)}$, is constructed such that $(X^{(2)})^\top X^{(2)} \approx ((X^{(1)})^\top X^{(1)})^{-2}$, and has labels $y_2 = 0$. Then, for balance parameter $\gamma$ sufficiently small, the AGOP second derivative approximately satisfies,

$$\mathbb{E}\left[\dot{W}^\top K \dot{W}\right] \sim X^\top Y X X^\top X X^\top Y X = (X^{(1)})^\top X^{(1)} X^\top X (X^{(1)})^\top X^{(1)} \approx (X^{(1)})^\top X^{(1)} (X^{(2)})^\top X^{(2)} (X^{(1)})^\top X^{(1)}$$
$$\approx I \ ,$$

In contrast, the NFM second derivative, $\mathbb{E}\left[\dot{W}^\top \dot{W}\right] = (X^\top Y X)^2 = ((X^{(1)})^\top X^{(1)})^2 \approx \Sigma^2$, will be significantly far from identity.

Specifically, we construct $X^{(2)}$ by the following procedure:

1. Extract singular values $S_1$ and right singular vectors $U_1$ from a singular-value decomposition (SVD) of $(X^{(1)})^\top X^{(1)}$.

2. Extract the left singular vectors $V_2$ from a sample $\tilde{X}_2$ that is sampled from the same distribution as $X^{(1)}$.

3. Construct $X^{(2)} = V_2 S_1^{-1} U_1^\top$.

4. Where $X = X^{(1)} \oplus X^{(2)}$, Set $X \leftarrow X + \epsilon_2 Z$, where $Z \sim \mathcal{N}(0, I)$.

5. Set $y \leftarrow y + 10^{-5} \cdot \underline{1}$.

Note that $U_1 S_1^{-1} V_2^\top V_2 S_1^{-1} U_1^\top = U_1 S_1^{-2} U_1^\top = ((X^{(1)})^\top X^{(1)})^{-2}$, therefore, we should set $X^{(2)} = V_2 S_1^{-1} U_1^\top$ to get $(X^{(2)})^\top X^{(2)} = ((X^{(1)})^\top X^{(1)})^{-2}$. Regarding the variance parameters, in practice we set $\epsilon_1 = 0.5$ and $\epsilon_2 = 10^{-2}$.

**Proposition 16** (Expected NFM and AGOP). *For a one hidden layer quadratic network, $f(x) = a^\top(Wx)^2$, with $a \sim \mathcal{N}(0, I)$ and $W \sim \frac{1}{\sqrt{k}} \cdot \mathcal{N}(0, I)$,*

$$\mathbb{E}_{a,W}\left[\dot{W}^\top \dot{W}\right] = (X^\top Y X)^2 \ ,$$

*and,*

$$\mathbb{E}_{a,W}\left[\dot{W}^\top K \dot{W}\right] = 3 \cdot tr\left(X^\top X\right) \cdot (X^\top Y X)^2$$
$$+ 6 X^\top Y X X^\top X X^\top Y X$$

*Proof of Proposition 16.*

$$\mathbb{E}\left[\dot{W}^\top \dot{W}\right] = X^\top Y X \mathbb{E}\left[W_0^\top \operatorname{diag}(a)^2 W_0\right] X^\top Y X$$
$$= (X^\top Y X)^2 \ .$$

Further,

$$\mathbb{E}\left[\dot{W}^\top K \dot{W}\right] = X^\top Y \mathbb{E}\left[K^2\right] Y X \ .$$

We note that,

$$K^2 = W_0^\top \operatorname{diag}(a)^2 W_0 X^\top X W_0^\top \operatorname{diag}(a)^2 W_0 \tag{29}$$

$$= \sum_{s_1,s_2}^{k} \sum_{\alpha}^{n} \sum_{p_1,p_2}^{d} \tag{30}$$

$$a_{s_1}^2 a_{s_2}^2 W_{s_1,p_1} X_{\alpha,p_1} X_{\alpha,p_2} W_{s_2,p_2} X W_{s_1} W_{s_2}^\top X^\top \ . \tag{31}$$

Therefore, applying Wick's theorem, element $i, j$ of $K^2$ satisfies,

$$\mathbb{E}\left[K_{ij}^2\right] = \sum_s^k \sum_\alpha^n \sum_{p_1,p_2}^d \mathbb{E}\left[a_s^4 W_{s,p_1} X_{\alpha,p_1} X_{\alpha,p_2} W_{s,p_2} X_i^\top W_s W_s^\top X_j\right] = \sum_s^k \sum_\alpha^n \sum_{p_1,p_2,q_1,q_2}^d$$

$$\mathbb{E}\left[a_s^4 W_{s,p_1} W_{s,p_2} W_{s,q_1} W_{s,q_2} X_{\alpha,p_1} X_{\alpha,p_2} X_{i,q_1} X_{j,q_2}\right]$$

$$= 3 \sum_s^k \sum_\alpha^n \sum_{p_1,p_2,q_1,q_2}^d$$

$$\left(\mathbb{E}\left[W_{s,p_1} W_{s,p_2}\right] \mathbb{E}\left[W_{s,q_1} W_{s,q_2}\right] + \right.$$

$$\mathbb{E}\left[W_{s,p_1} W_{s,q_1}\right] \mathbb{E}\left[W_{s,p_2} W_{s,p_2}\right] + $$

$$\left. \mathbb{E}\left[W_{s,p_1} W_{s,q_2}\right] \mathbb{E}\left[W_{s,p_2} W_{s,q_1}\right]\right)$$

$$\cdot X_{\alpha,p_1} X_{\alpha,p_2} X_{i,q_1} X_{j,q_2}$$

$$= 3 \sum_\alpha^n \sum_{p_1,p_2,q_1,q_2}^d$$

$$\left(\delta_{p_1 p_2} \delta_{q_1 q_2} + \delta_{p_1 q_1} \delta_{p_2 q_2} + \delta_{p_1 q_2} \delta_{p_2 q_1}\right)$$

$$\cdot X_{\alpha,p_1} X_{\alpha,p_2} X_{i,q_1} X_{j,q_2}$$

$$= 3 \sum_\alpha^n \left(\sum_{p_1,q_1}^d X_{\alpha,p_1} X_{\alpha,p_1} X_{i,q_1} X_{j,q_1}\right.$$

$$+ \sum_{p_1,p_2}^d X_{\alpha,p_1} X_{\alpha,p_2} X_{i,p_1} X_{j,p_2}$$

$$\left. + \sum_{p_1,p_2}^d X_{\alpha,p_1} X_{\alpha,p_2} X_{i,p_2} X_{j,p_1}\right)$$

$$= 3 \cdot \operatorname{tr}\left(X^\top X\right) \cdot X_i^\top X_j + 3 \sum_\alpha^n X_\alpha^\top X_i X_\alpha^\top X_j$$

$$+ 3 \sum_\alpha^n X_\alpha^\top X_j X_\alpha^\top X_i$$

$$= 3 \cdot \operatorname{tr}\left(X^\top X\right) \cdot X_i^\top X_j + 3 X_i X^\top X X_j + 3 X_j X^\top X X_i \ .$$

Finally, we conclude,

$$\mathbb{E}\left[K^2\right] = 3\left(\operatorname{tr}\left(X^\top X\right) X X^\top + 2 X X^\top X X^\top\right) \ ,$$

giving the second statement of the proposition. $\qquad \square$

## H   Varying the data distribution

We verify that our observations for isotropic Gaussian data hold even when the data covariance has a significant spectral decay. (Figures 9 and 10). We again consider Gaussian data that is mean 0 and where the covariance is constructed from a random eigenbasis. In Figure 9, we substitute the eignevalue decay as $\lambda_k \sim \frac{1}{1+k}$, while in Figure 10, we use $\lambda_k \sim \frac{1}{1+k^2}$. We plot the values of the UC-NFC, C-NFC, train loss, and test loss throughout training for the first and second layer of a two hidden layer network with ReLU activations, while additionally varying initialization scale. Similar to Figure 8, we observe that the C-NFC is more robust to the initialization scale than the UC-NFC, and UC-NFC value become high through training, while being small at initialization. We see that the test loss improves for smaller initializations, where the value of the C-NFC and UC-NFC are higher.

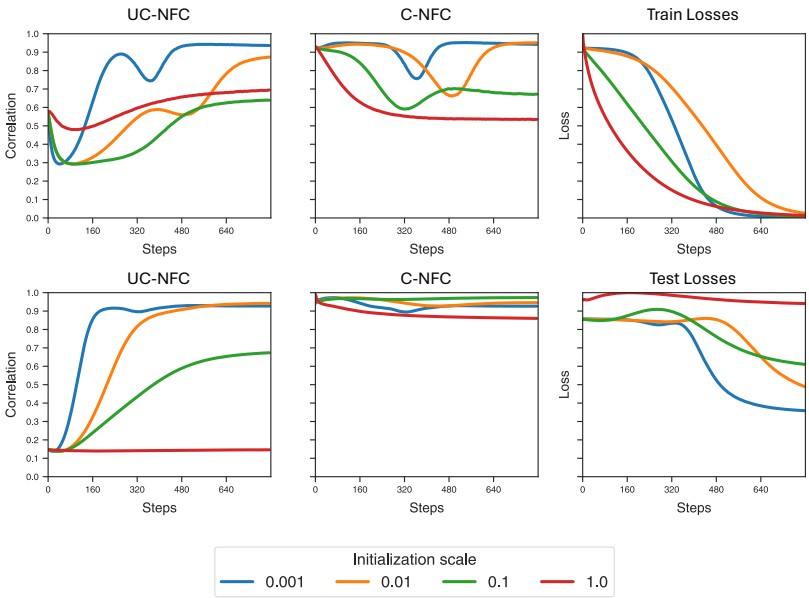

Figure 8: Centered neural feature correlations. Data covariance decay rate $\lambda_k = 1$. Top row is layer 1, bottom row is layer 2. Train (test) losses are scaled by the maximum train (test) loss achieved so that they are between 0 and 1.

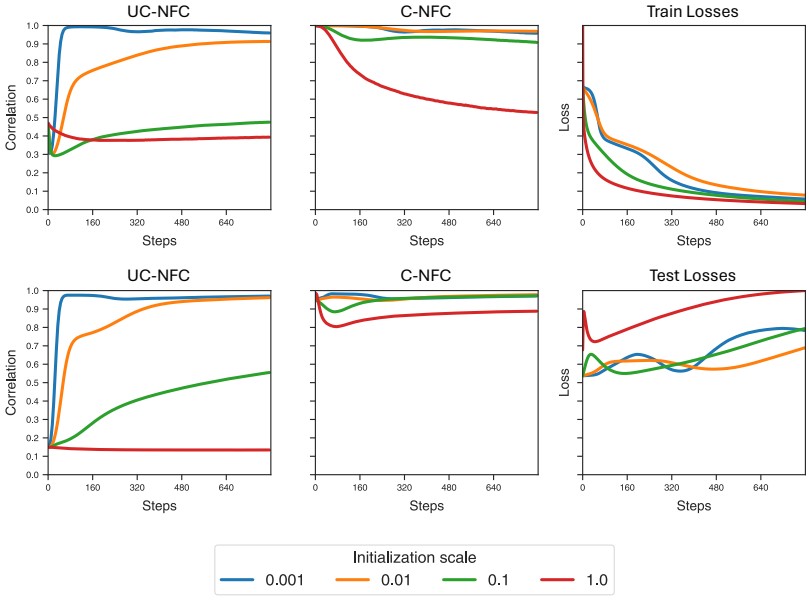

Figure 9: Centered neural feature correlations. Data covariance decay rate $\lambda_k \sim \frac{1}{1+k}$. Top row is layer 1, bottom row is layer 2. Train (test) losses are scaled by the maximum train (test) loss achieved so that they are between 0 and 1.

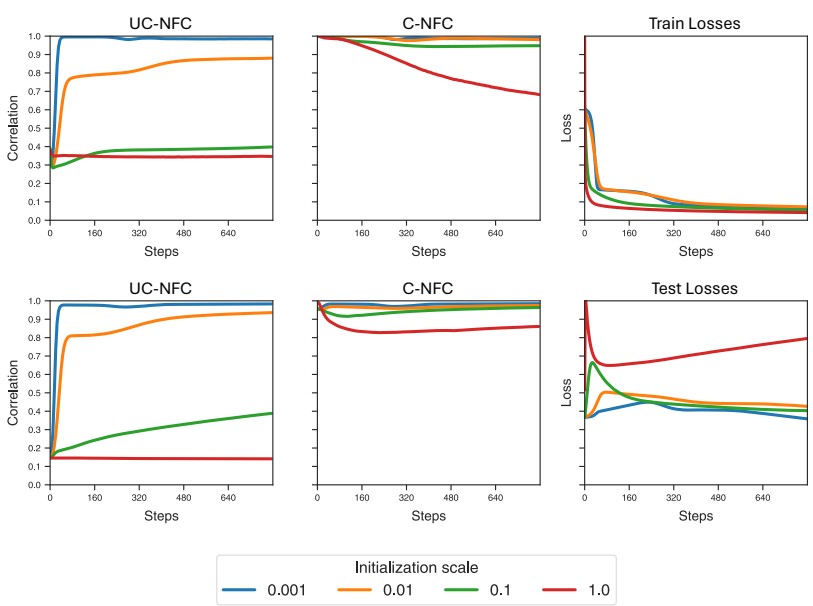

Figure 10: Centered neural feature correlations. Data covariance decay rate $\lambda_k \sim \frac{1}{1+k^2}$. Top row is layer 1, bottom row is layer 2. Train (test) losses are scaled by the maximum train (test) loss achieved so that they are between 0 and 1.

# I Effect of initialization on feature learning

We see that when initialization is small, the C-NFC and UC-NFC are high at the end of training with and without fixing the learning speed (Figure 4). This is reflected by the quality of the features learned by the NFM and the qualitative similarity of the NFM and AGOP at small initialization scale (Figure 11). Further, we notice that as we increase initialization, without fixing speeds, the correspondence between the NFM and decreases and the quality of the NFM features decreases (at a faster rate than the AGOP). Strikingly, when learning speeds are fixed, the quality of the features in the AGOP and NFM becomes invariant to the initialization scale.

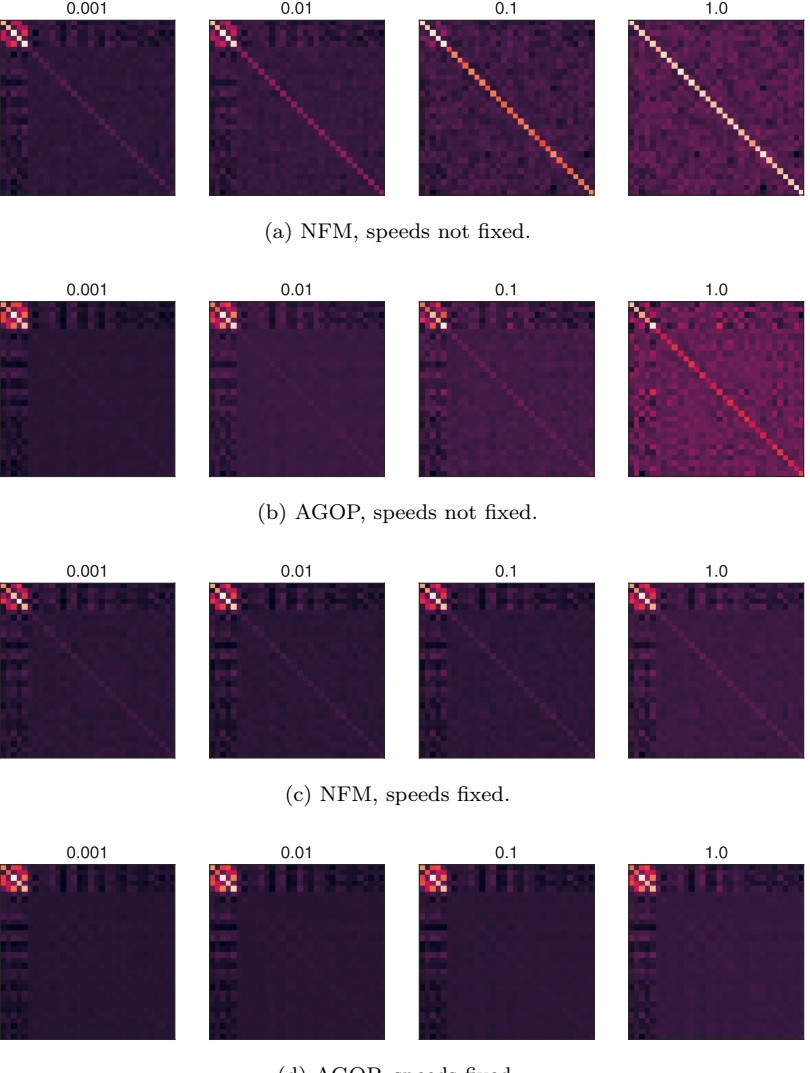

Figure 11: The NFM at the end of training as a function of initialization scale in the first layer weights, with and without fixing learning speeds. The task again is the chain monomial with rank $r = 5$. The title of each plot is the initialization scale of the first layer $s_0$.

## J   Experimental details

We describe the neural network training and architectural hyperparameters in the experiments of this paper. Biases were not used for any networks. Further, in all polynomial tasks, we scaled the label vector to have standard deviation 1.

**Corrupted AGOP**   For the experiments in Figure 1, we used $n = 384$ data points, $d = 32$, $k = 128$ as the width in all layers, isotropic Gaussian data, initialization scale 0.01 in the first layer and default scale in the second. We used ReLU activations and two hidden layers. For the experiments in Figure 8,9,10,5, and 6, we used a two hidden layer network with ReLU activations, learning rate 0.05, 800 steps of gradient decent, and took correlation/covariance measurements every 5 steps.

**C/UC-NFC calculations on real datasets**   We describe the experimental details for Figure 2. For (A,B) we trained a five layer MLP on the first 50,000 datapoints of *Streetview House Numbers* (SVHN), CIFAR-10, CIFAR-100, STL-10, MNIST, and *German Traffic Sign Recognition Benchmark* (GTSRB) datasets. We used the default PyTorch initialization (scale of 1) for all layers. We used width 256 in all layers, and trained with SGD with batch size 128. For SVHN and CIFAR, we trained for 150 epochs with learning rate 0.2. For STL-10 and GTSRB we used learning rate 0.1 for 150 epochs. For MNIST we trained for 50 epochs with learning rate 10. For the VGG-11 experiments on CIFAR-10, we used the default architecture from `torchvision` with batch-norm layers removed. We trained for 50 epochs and learning rate 1. For experiments with a GPT-family model, we adapted the model and dataset from NanoGPT (`https://github.com/karpathy/nanoGPT`). We used all the default settings and the default Adam optimizer, but used no weight decay, learning rate 5e-3, and removed all dropout layers. We also reduced the number of attention layers to 3 from the original 6. We trained on the Shakespeare characters dataset.

**Alignment reversing dataset**   For the experiments in Figure 3, we used $k = n = d = 1024$ for the width, dataset size, and input dimension, respectively. Further, the traces of powers of $F_a$ are averaged over 30 neural net seeds to decouple these calculated values from the individual neural net seeds. The mean value plotted in the first two squares of figure is computed over 10 data seeds.

**SLO experiments**   For the SLO figures (Figures 11, 4), we use isotropic Gaussian data, 600 steps of gradient descent. The learning rates are chosen based on initialization scale in the first layer. For initialization scales 1, 0.1, 0.01, and 0.001, we used learning rates 0.03, 0.1, 0.2, 0.4, respectively. We again used two hidden layers with ReLU activations. We chose $n = 256$, $d = 32$, and $k = 256$ as the width. We divided the linear readout weights by 0.01 at initialization to promote feature learning, and modified SLO to scale gradients by $(\epsilon + \|\nabla \mathcal{L}\|)^{-1}$, rather than just the inverse of the norm of the gradient, for $\epsilon = 0.1$. This technique smooths the training dynamics as the parameters approach a loss minimum, allowing the network to interpolate the labels.

**Predictions with depth**   For the Deep C-NFC predictions (Figure 7), we used $n = 128$, $d = 128$, initialization scale of 1. The low rank task is just the chain monomial of rank $r = 5$. The high rank polynomial task is $y(x) = \sum_{i=1}^{d}(Qx)_i^2$, where $Q \in \mathbb{R}^{d \times d}$ is a matrix with standard normal entries.

**Figures 16 and 17**   For the experiments on the SVHN dataset, we train a four hidden layer neural network with ReLU activations, initialization scale 1.0 in all layers and width 256. For SVHN, we subset the dataset to 4000 points. We train for 3000 epochs with learning rate 0.2 for standard training, and 0.3 for SLO, and take NFC measurements every 50 epochs. For SLO, we set $C_0 = 2.5$, $C_1 = C_2 = 0.4$, and relaxation parameter $\epsilon = 0.2$. We pre-process the dataset so that each pixel is mean 0 and standard deviation 1. For the experiments on CelebA, we train a two hidden layer network on a balanced subset of 7500 points with Adam with learning rate 0.0001 and no weight decay. We use initialization scale 0.02 in the first layer, and width 128. We train for 500 epochs. We pre-process the dataset by scaling the pixel values to be between 0 and 1.

**Code availability**   We make the code available for the experiments in Figure 2 available at `https://anonymous.4open.science/r/centered_NFA-4795/` (for MLP+VGG).

# K   Additional SLO experiments

We demonstrate the SLO can be applied adaptively to increase the strength of the UC-NFC in all layers of a deep network on the chain monomial task of rank $r = 3$. We train a three hidden layer MLP with ReLU activations and an initialization scale of 0.1 by SLO, and find that all layers finish at the same high UC-NFC (Figure 12). Further, this final UC-NFC value is higher than the highest UC-NFC achieved by any layer with standard training. The generalization loss is also lower with SLO on this example, corresponding to better feature learning (through the UC-NFC).

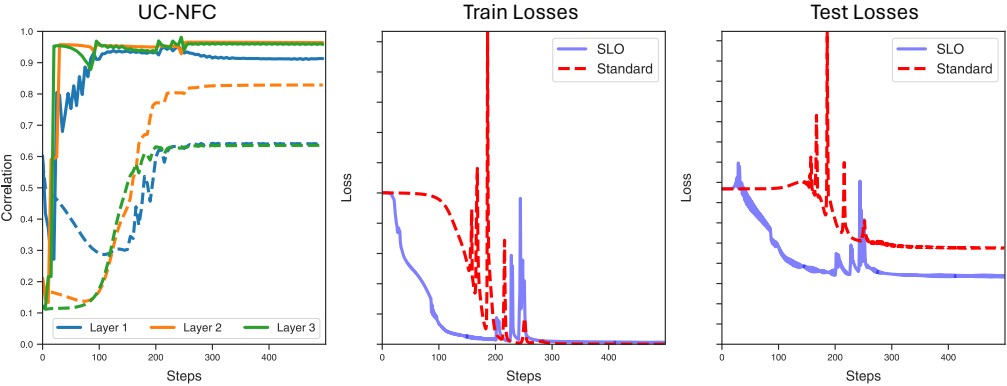

Figure 12: Training with SLO where the learning speeds are chosen adaptively based on the UC-NFC values of all layers. The dashed lines correspond to training with standard GD.

At every time step we choose $C_i = s$ for the layer $i$ with the smallest UC-NFC correlation value, while setting $C_j = s^{-1}$ for all other layers, with $s = 20$. We again modify SLO by dividing the gradients by $\epsilon \|\nabla \mathcal{L}\|$ for $\epsilon = 0.01$. The learning rate is set to 0.05 in SLO and 0.25 for the standard training (gradient descent), and the networks are trained for 500 epochs. We sample $n = 256$ points with $d = 32$, and use width $k = 256$.

## L    Additional C-NFC/UC-NFC plots across architectures

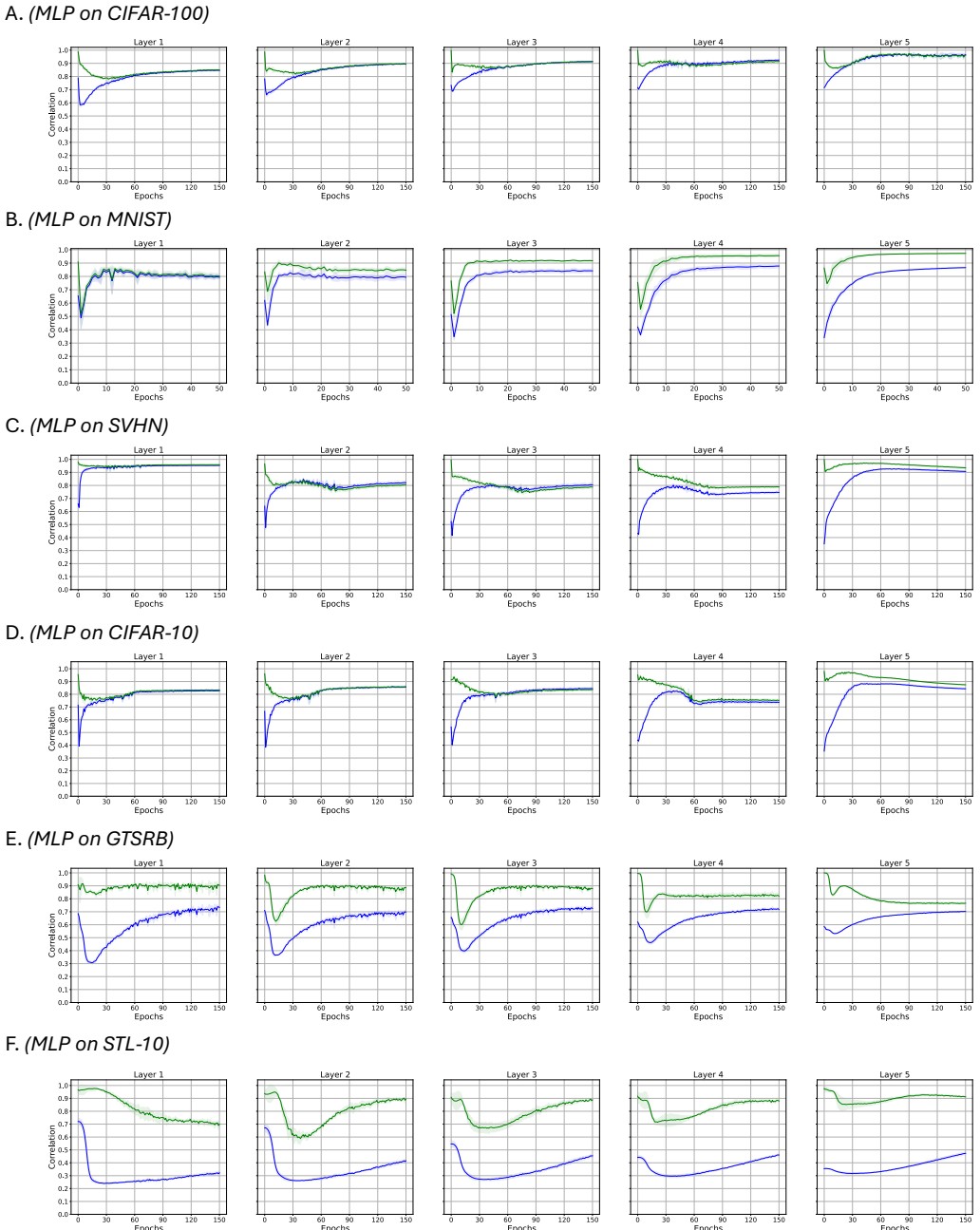

Figure 13: Full trajectories for the C/UC-NFC of a five hidden layer MLP trained on six datasets, averaged over three seeds, with large (1.0) initialization scale. The blue curves are the UC-NFC, while green curves are the C-NFC.

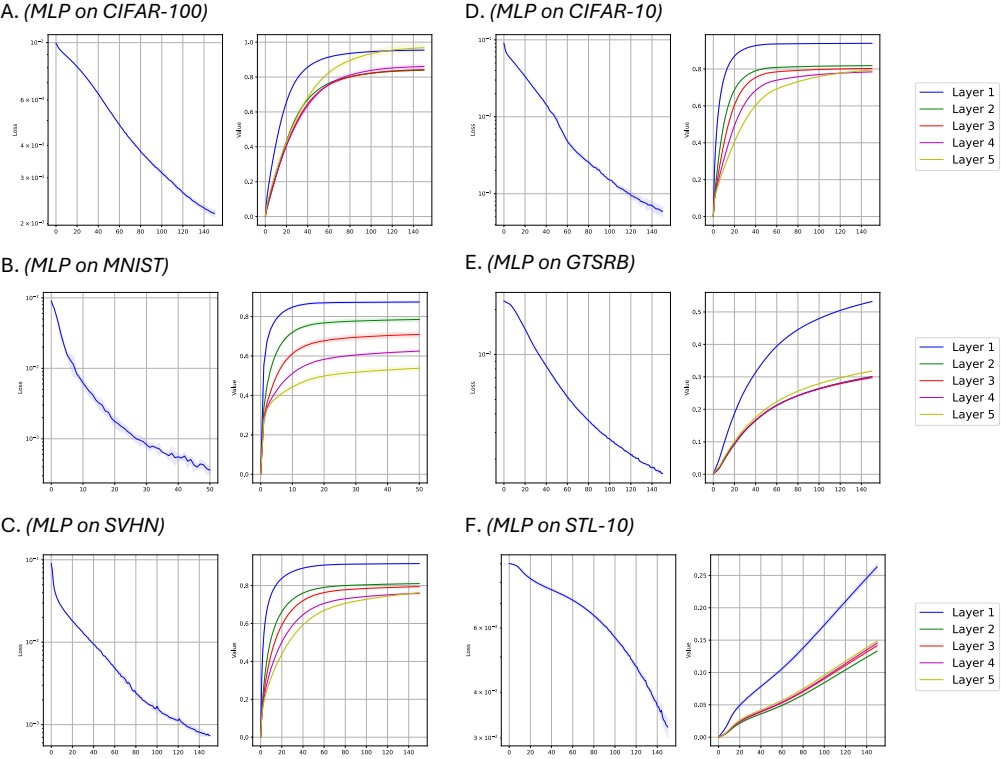

Figure 14: Losses and normalized changes in weights across datasets for a five hidden layer MLP. The change in weight is measured as $\|W - W_0\|\|W\|^{-1}$. First column of all subfigures are the losses, while the second columns are the weight changes.

## M    Additional experiments on real datasets

We replicate Figures 1 and 4 on celebrity faces (CelebA) and Street View House Numbers (SVHN). We begin by showing that one can disrupt the NFC correspondence by replacing the PTK feature covariance with a random matrix of the same spectral decay. For this example, we measure the Pearson correlation, which subtracts the mean of the image. I.e. $\bar{\rho}(A, B) \equiv \rho\left(A - m(A), B - m(B)\right)$, where $m(A), m(B)$ are the average of the elements of $A$ and $B$.

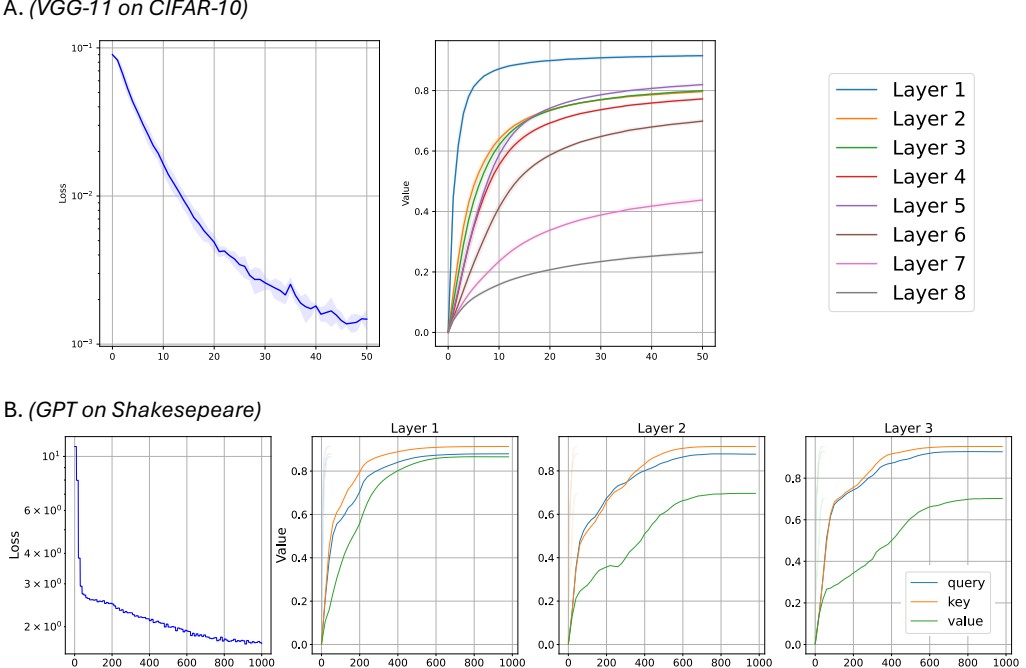

Figure 15: Losses and normalized changes in weights across datasets for VGG-11 on CIFAR-10 and GPT on Shakespeare character generation. The change in weight is measured as $\|W - W_0\| \|W\|^{-1}$.

First column of (A) and (B) are the losses, while the remaining columns are the weight changes.

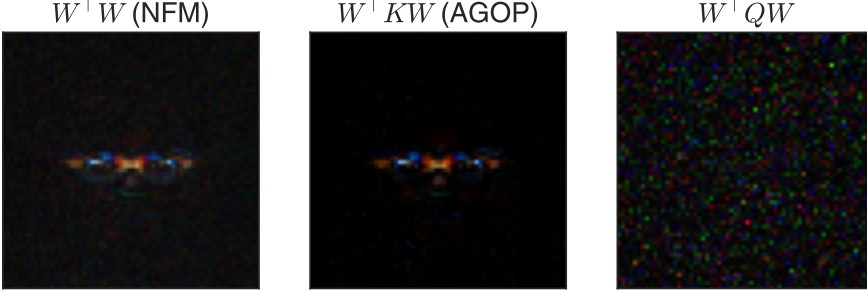

Figure 16: Various feature learning measures for the CelebA binary subtask of predicting glasses. The diagonals of the NFM $\left(W^\top W\right)$ (first plot) and AGOP $\left(W^\top K W\right)$ (second plot) of a fully-connected network are similar to each other. Replacing $K$ with a symmetric matrix $Q$ with the same spectrum but independent eigenvectors obscures the low rank structure (third plot), and reduces the Pearson correlation of the diagonal from $\bar{\rho}\left(\text{diag}\left(F\right), \text{diag}\left(\bar{G}\right)\right) = 0.91$ to $\bar{\rho}\left(\text{diag}\left(F\right), \text{diag}\left(W^\top Q W\right)\right) = 0.04$.

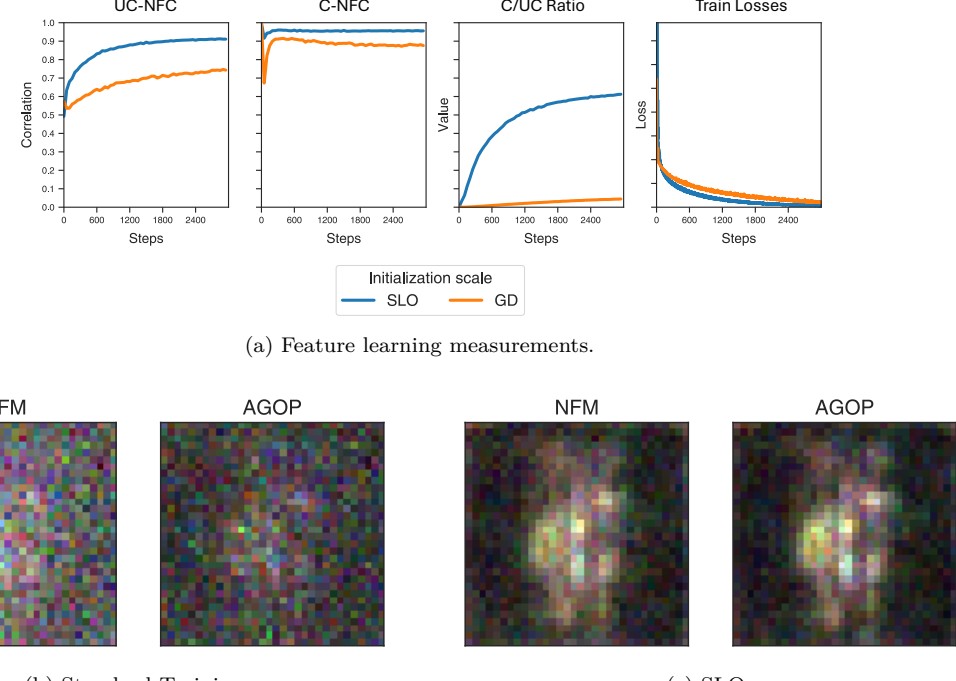

(a) Feature learning measurements.

(b) Standard Training          (c) SLO

Figure 17: We demonstrate on the SVHN dataset, with a 4 hidden layer neural network with large initialization scale, how SLO can improve the strength of the UC-NFC, the C-NFC, the ratio of the unnormalized C-NFC to UC-NFC (plot (a)) and the feature quality (plots (b) and (c)). In plots (b) and (c), we visualize the diagonal of the NFM and AGOP for the first layer of the trained network, where SLO was applied with $C_0 = 2.5$, $C_1 = C_2 = 0.4$.

