# OpenReview forum: "Feature learning as alignment: a structural property of gradient descent in non-linear neural networks"
_TMLR — Accepted by TMLR_

### Review · Reviewer_UL1M · 2024-08-23

**Summary Of Contributions:**

The authors claim to provide both an experimental and theoretical study of the Neural Feature Ansatz (NFA), namely that the Gram Matrix of each layer aligns during training to the Average gradient outer product matrix (AGOP). They provide experiments for different architectures/dataset and give theoretical ground for this phenomenon in the case of gradient flow training. Finally, the authors claim to provide an algorithm that boosts the NFA phenomenon in order to do more feature learning.

**Audience:**

Yes

**Broader Impact Concerns:**

I have no concern.

**Claims And Evidence:**

No

**Requested Changes:**

*Major changes*

My main change request is either to forget that the paper is something more than experimental or to largely change the theoretical part: being clear about the assumption, the statement etc...

*Minor changes*

- The discussion on the necessity to deal with the centered and uncentered versions of the feature covariance matrices is rather confusing. I would like the authors to clean this point: notably, what is the link with small initialization on this regard?

**Strengths And Weaknesses:**

**Strengths**

Here is a list of the strengths of the paper, according to me:

- The authors try to tackle an important question : *how a neural network learn features adapted to the data*. To do this, they try to follow the NFA and provide complementing experiments and phenomenology to it. These questions are very important to understand Neural Network training.

- The paper -or at least the story line- is quite clear and easy to follow.

- The authors present many experiments that show the phenomenon they study

*Weaknesses**

Here is a list of the weaknesses of the paper, according to me:

- While the notations and concept are well introduced, and the experiments conclusive, the theoretical contributions trying to explain the phenomenon described are questionable are rather weak compared to the claims the authors make. Here are few important examples of this:
  - *On Proposition 3*: the mathematical hypothesis are not properly introduced, for example why is it said that $\bar{W} = 0$, two lines before ? I guess that this corresponds to the fact that all the derivatives in the proposition are taken at $t = 0$ (right after initialization), but it took me quite some time to understand it. Moreover, why the authors claim that *the first non-trivial time derivatives are highly correlated*? There is a $\mathcal{K}$ vs $\mathcal{K}^2$ difference between both derivatives and I do not understand how this can be benign. Finally, the alignment is only valid locally at $t = 0$, at second order, this is far from being enough to conclude.
  - Some mathematical statement are rather rapid, e.g. equations $(6)$ then $(7)$ are provided under certain assumptions, and this is never *clearly stated*. Moreover, from these, conclusion are given as if it were general cases. Another striking example is on the fact that there is a validation of the results in Section 4.2,  under very specific hypothesis (one layer, quadratic activation, $X^\top X = I$...)

---

> ### Author Response · Authors · 2024-09-18
> **Author Response**
>
> We thank the reviewer for their review. We first address their broad concern:
>
> **“My main change request is either to forget that the paper is something more than experimental or to largely change the theoretical part: being clear about the assumption, the statement etc…”**
>
> In order to address these concerns, we have made the assumptions and limitations of our theoretical setting more explicit. In addition, we have a new theoretical result (Section 4.2) based on a suggestion from reviewer VtKo with detailed description of assumptions. With these changes we believe that the theoretical part of our paper is valid and appropriately scoped.
>
>
> _Responses to the individual points:_
>
> **“On Proposition 3: the mathematical hypothesis are not properly introduced, for example why is it said that $\bar{W} = 0$, two lines before? I guess that this corresponds to the fact that all the derivatives in the proposition are taken at $t = 0$ (right after initialization), but it took me quite some time to understand it.”**
>
> We made it more clear why $\bar{W}=0$ in our revised manuscript.
>
> **“Moreover, why the authors claim that the first non-trivial time derivatives are highly correlated?”**
>
> Per our global comment, we formally provide a general case where the centered derivatives are perfectly correlated asymptotically.
>
> **“Some mathematical statement are rather rapid, e.g. equations $(6)$ then $(7)$ are provided under certain assumptions, and this is never clearly stated.”**
>
> We appreciate this critique. Equations 6 and 7 are in fact exact; the first term in the right hand side of equation 6 is equal to the left hand side of equation 7 in a high dimensional limit which we now note explicitly in the main text. We also have an additional theoretical result in a different setting which can be found in Section 4.2, alongside its key assumptions.
>
> **“Moreover, from these, conclusion are given as if it were general cases. Another striking example is on the fact that there is a validation of the results in Section 4.2, under very specific hypothesis (one layer, quadratic activation, $X^\top X = I$...)”**
>
> The case of $X^\top X = I$ was presented for exposition, but we give calculations for the general case. We left extension of the analytical predictions to deeper networks with alternative activation functions to future work due to the complexity of the analysis - the total number of terms can grow rapidly for activations with non-zero mean, and with depth. Our experiments on real networks suggest that qualitatively similar results will hold in these settings.
>
> **“The discussion on the necessity to deal with the centered and uncentered versions of the feature covariance matrices is rather confusing. I would like the authors to clean this point: notably, what is the link with small initialization on this regard?”**
>
> We introduce the centered quantities to understand whether the updates to the weight matrices directly drive alignment with the PTK feature covariance, or whether the alignment is driven by alignment of the PTK to the initial weights. High values for the centered NFC suggest the former effect is responsible for the NFA. We clarify the role of small initialization in Section 4 of our revised manuscript. Namely, small initialization will eliminate the additional terms in the UC-NFC from the initial weights later in training, causing the C-NFC and UC-NFC to coincide. This is because $\bar{W} = W - W_0 \rightarrow W$, as the initialization scale goes to zero.

---

> > ### Comment · Reviewer_UL1M · 2024-09-29
> > **After review**
> >
> > I have read the answer of the authors, that globally argue that they have detailed all the precision that I have asked. This is welcome. I have no time to read through entirely the change made by the authors, but this appears convincing.
> >
> > ------------------------
> >
> > After rebuttal, due to the imprecisions of the first draft and the induced fuzziness of the conclusions driven by the authors, I prefer to stay neutral w.r.t. the paper acceptance. The two other reviewers seem convinced by the paper and I would be okay for an acceptance if they still want it to be accepted.

---

### Review · Reviewer_VtKo · 2024-09-04

**Summary Of Contributions:**

This submission studies, both theoretically and empirically, the neural feature ansatz (NFA) proposed in (Radhakrishnan et al. 2024), which states that feature learning results in the Gram matrix of a neural network layer aligning with the average gradient outer product (AGOP) with respect to the input to that layer. The authors identified the alignment between the weight matrix and the preactivation tangent kernel matrix as an indicator of NFA and introduced the centered neural feature correlation (C-NFC) to measure such alignment. The authors also provided some theoretical justification on how NFA may arise in the early stages of gradient descent, and proposed a gradient-based update rule to increase the strength of C-NFC and hence feature learning.

**Audience:**

Yes

**Claims And Evidence:**

Yes

**Requested Changes:**

Please refer to the weakness section above.

**Strengths And Weaknesses:**

The mechanism of feature learning in neural networks is an important research question. This submission presents a mix of empirical and theoretical investigations of a fairly general feature learning mechanism that goes beyond the idealized problem settings in prior theoretical works, which I believe is a valuable contribution.

I have the following concerns/questions:

1. The analysis of gradient descent dynamics is limited, and the connection between C-NFC and learning needs to be elaborated.

- The current gradient descent analysis is basically restricted to the first gradient step, as it does not account for the parameters depending on the training data. Due to the simplified setting, my impression is that it should be possible to analytically compute the NFC in some idealized student-teacher settings (for the two-layer model at least), using similar Gaussian equivalence calculations similar to (Ba et al. 2022).  Can the authors comment on the technical challenges here?

- The authors wrote: "*The relation between the NFM and the AGOP is significant, in part, because for a neural network that has learned enough information about the target function, the AGOP of this model with respect to the first-layer inputs will approximate the expected gradient outer product (EGOP) of the target function.*"
Can the authors explain what kind of information is contained in the EGOP, and why it is "enough" for learning? The experiment in Figure 1 seems to suggest that this matrix encodes the support of the target function; how does this relate to the low-rank support recovery mechanism in (Damian et al. 2022) (Abbe et al. 2022)?

2. The presentation of theoretical results (especially in Section 4.1) can be improved.

- Can the authors clarify what is the "*appropriate high-dimensional limit*" in Section 4.1? Under what scaling of data size and model width do the authors expect the asymptotic freeness to hold?

- The eigenstructure of $\mathcal{K}$ at initialization is not explicitly characterized, and there are no general principles that decide the extent of alignment between the derivatives in Proposition 3, even in the simple quadratic case.

3. A few related works are missing.

- How gradient descent induces alignment between the target function and certain feature matrices has been studied in the random matrix theory literature. For example, (Ben Arous et al. 2023) analyzed the Hessian and (Wang et al. 2024) the NFM. It would be a good idea to discuss whether these rigorous analyses apply to the problem setting in Section 4.1.
(Ben Arous et al. 2023) *High-dimensional SGD aligns with emerging outlier eigenspaces*.
(Wang et al. 2024) *Nonlinear spiked covariance matrices and signal propagation in deep neural networks*.

- How does the speed-limited optimization relate to the feature learning parameterizations in (Yang and Hu 2022) (Chizat and Netrapalli 2023)? Both works enforced the parameters of the first layer to move faster.
(Yang and Hu 2022) *Feature learning in infinite-width neural networks*.
(Chizat and Netrapalli 2023) *The Feature Speed Formula: a flexible approach to scale hyper-parameters of deep neural networks*.

---

> ### Author Response · Authors · 2024-09-18
> **Author Response Part 1**
>
> We thank the reviewer for their review. We respond to the points raised.
>
> **“The current gradient descent analysis is basically restricted to the first gradient step, as it does not account for the parameters depending on the training data. Due to the simplified setting, my impression is that it should be possible to analytically compute the NFC in some idealized student-teacher settings”**
>
> Per our global comment, we formally provide a case where the centered derivatives are perfectly correlated asymptotically (Section 4.2 in the revision). Our proof uses a version of the Gaussian equivalence suggested - namely that kernel matrices can be well-approximated by the first terms in the Taylor expansion in the limit that $n,d \rightarrow \infty$ together.
>
> **“Can the authors explain what kind of information is contained in the EGOP, and why it is "enough" for learning? The experiment in Figure 1 seems to suggest that this matrix encodes the support of the target function; how does this relate to the low-rank support recovery mechanism in (Damian et al. 2022) (Abbe et al. 2022)?”**
>
> The EGOP will enable learning functions with low-rank support that are otherwise unlearnable by fixed kernel methods (see e.g. [1]). Damian et al. demonstrate that one step of gradient descent is sufficient to learn this support when the number of samples is super-quadratic in the data dimension, though typically in practice one step is not sufficient. Abbe et al. mainly consider the setting where neural networks trained with online SGD can learn the support with $O(d)$ samples by hierarchically learning one component at a time. These works analyze the obtained structure by studying different statistics than the EGOP.
>
> We note that while the EGOP is motivated by learning this support, the EGOP can contain more general structure of the target function that is not necessarily low rank. For example, the EGOP has enabled learning modular arithmetic with full rank features [2].
>
> [1] Ghorbani, Mei, Misiakiewicz, Montanari, “When do neural networks outperform kernel methods?”, NeurIPS 2020.
>
> [2] Mallinar, Beaglehole, Zhu, Radhakrishnan, Pandit, Belkin, “Emergence in non-neural models: grokking modular arithmetic via average gradient outer product”, arXiv preprint 2024.
>
> **“Can the authors clarify what is the "appropriate high-dimensional limit" in Section 4.1?”**
>
> The high dimensional limit we are interested in this section is linearly co-scaling input dimension, hidden dimension, and number of datapoints. We assume additional free independence of $X$ and $Y$ with the initial parameters. See our revised manuscript for additional discussion of our assumptions here.
>
> **“The eigenstructure of $\mathcal{K}$ at initialization is not explicitly characterized, and there are no general principles that decide the extent of alignment between the derivatives in Proposition 3, even in the simple quadratic case.”**
>
> As per our response in point 1, we have proven that the alignment will be exact for these derivatives for data uniformly distributed on the sphere in $d$ dimensions.

---

> > ### Author Response · Authors · 2024-09-18
> > **Author Response Part 2**
> >
> > **“A few related works are missing”**
> >
> > Thank you for pointing out these works. We will provide references to these works in the updated manuscript. We address how each might be relevant to our work here:
> > 1. (Ben Arous et al. 2023) High-dimensional SGD aligns with emerging outlier eigenspaces.
> >
> > This work demonstrates in several particular settings that the weight trajectories align with the Hessian and gradient second moment matrices of the loss function with respect to the parameters, measured on the test data. These analyses do not directly apply to our setting, as we consider derivatives with respect to the input data and activations. It would be interesting to determine if our notions of alignment can be rigorously related to one another.
> >
> > 2. (Wang et al. 2024) Nonlinear spiked covariance matrices and signal propagation in deep neural networks.
> >
> > This reference studies how spikes in the data covariance, potentially induced through feature learning, can propagate to the conjugate kernel of the neural network. This provides a promising avenue to study the time-evolution of the PTK matrix through (at least) early periods of training.
> >
> > 3. (Yang and Hu 2022) Feature learning in infinite-width neural networks.
> >
> > We agree with your interpretation of the relationship between this work and Speed Limited Optimization (SLO). Intuitively, by reducing the scale of the last layer weights, the first layer weights are required to move significantly (in $\ell_\infty$ norm) to fit the data. The SLO intervention is more extreme in that it overrides the natural GD dynamics to exactly set the weight movement rates (what we call ``learning speeds’’) per layer.
> >
> > 4. (Chizat and Netrapalli 2023) The Feature Speed Formula: a flexible approach to scale hyper-parameters of deep neural networks.
> >
> > This work considers the learning speeds with respect to the activations at every layer, and not the weights themselves. Nonetheless, this work is quite related because they adjust learning rates in each layer so as to maintain large learning speeds, according to their notion, across all layers. Instead, SLO sets learning speeds of the weights in each layer so that some weights move much faster than others, depending on the layer where we want to maximize feature learning.

---

### Review · Reviewer_wFmu · 2024-09-13

**Summary Of Contributions:**

The authors investigate a concrete structural property of neural networks which is hypothesized to be tightly linked with feature learning. Specifically they seek to the origins of the so called Neural Feature Ansatz (NFA). The NFA is that after training, the Gram matrix of the weights (Neural Feature Matrix or NFA) is highly correlated (as measured by the Neural Feature Correlation or NFC) with the Average Gradient Outer Products (AGOP). Firstly, to more tightly link the NFA and the AGOP the authors observe that the AGOP can be written in terms of the weight matrices and the Preactivation Kernel Matrix (PKM). The authors then use this framing to investigate the origins and implications of the NFA phenomenon, providing several observations:

1. The authors empirically demonstrate that at early times the NFA and AGOP alignment is driven by the alignment of the PKM with the *change* in the weights, which the authors call the centered neural feature correlation (C-NFC).

2. The authors analyze the dynamics of the NFA and AGOP in a simple setting at initialization to provide a heuristic argument for the emergence of the NFA and AGOP correlation. These calculations allow the authors to demonstrate that this emergence is not universal and that a carefully constructed dataset can break it.

3. The authors measure the correlation between the initialization scale and the NFC, showing empirically that higher correlation and better generalization occurs with smaller initialization scales which intuitively is linked with better feature learning.

4. Lastly, the authors use their observations to inspire a new optimization variant called Speed Limited Optimization (SLO). The authors argue and provide some experimental evidence that this optimization algorithm better promotes NFC and hence better feature learning and generalization.

**Audience:**

Yes

**Broader Impact Concerns:**

No concerns.

**Claims And Evidence:**

Yes

**Requested Changes:**

I believe the submission is already quite strong and deserving of acceptance. I would request the following changes to strengthen the work:

1. Provide some introduction and background on free probability for readers who are not already familiar. For example: "We make the assumption that X and Y are (asymptotically) freely independent of the parameters at initialization", does this just follow from classical independence?

2. In Eq. (7) why does the NTK limit imply that the Cov term goes to zero?

3. Can the theory be used to understand why the C-NFC is the primary contributor to NFC? Or how the initialization scale affects the correlation?

4. The $\eta$ in the SLO update should be a $-\eta$?

4. The motivation for SLO is unclear, as is the statement: "We expect this rule to increase
the strength of the UC-NFC in layers where $C_l$ is large relative to $C_m$ for $m \neq l$". Also the appropriate baseline should be to tune all layerwise learning rates, since each layer has a speed. Is there a reason to believe that against this baseline SLO should be better? Is there a way to appropriately set the layerwise speeds?

**Strengths And Weaknesses:**

Strengths: The paper is overall well-written and contains many interesting observations about NFA. Understanding feature learning in neural networks is very difficult and so conceptual properties such as the NFA are useful for making progress on this important topic. The origins of NFA have so far been quite mysterious. The experiments and derivations of this work are a positive contribution towards shedding light on the origins of NFA.

Weaknesses: Some of the explanations and claims feel a bit too hand-wavy. For example, the justification for the Speed Limited Optimization was not completely clear to me. I believe these can benefit from more clarifications. The theoretical setting is also fairly limited.

---

> ### Author Response · Authors · 2024-09-18
> **Author response**
>
> We thank the reviewer for their review. We respond to each of the requested changes.
>
> **“The theoretical setting is also fairly limited.”**
>
> Per our global comment, we formally provide a case where the centered derivatives are perfectly correlated in a well studied high-dimensional limit.
>
> **“Provide some introduction and background on free probability for readers who are not already familiar. For example: "We make the assumption that X and Y are (asymptotically) freely independent of the parameters at initialization", does this just follow from classical independence?”**
>
> Free probability is the non-commutative analog of classical independence — a standard method for understanding random matrices and their products. We have added additional discussion of our assumptions and references on this topic. In general, classical independence of the entries implies free independence of the matrices _asymptotically_ in the high dimensional limit.
>
> **“In Eq. (7) why does the NTK limit imply that the Cov term goes to zero?”**
>
> The covariance goes to $0$ because $\mathcal{K}$ is deterministic in this limit. Hence $\mathcal{K} = \mathbb{E}{\mathcal{K}}$ and $\mathcal{K^2} = \mathbb{E}{\mathcal{K}^2}$. Recall that $Cov(A,B) = \mathbb{E}[(A - \mathbb{E}(A))(B - \mathbb{E}(B))] for random variables $A,B$.
>
> **“Can the theory be used to understand why the C-NFC is the primary contributor to NFC? Or how the initialization scale affects the correlation?”**
>
> The initialization scale affects the correlation since the difference between the C-NFC and the NFC can be written in terms of the initialization. This is clear from the fact that $\bar{W} = W - W_0 \rightarrow W$ as the initialization scale goes to $0$ where $W$ is extracted at the end of training. Therefore small initialization leads the C-NFC and NFC to be similar at late times, regardless of other dynamics.
>
> In general, the C-NFC is the primary contributor to the NFC if either 1. The final weight matrices are much larger in magnitude than the initial weight matrices or 2. If the final weight matrices are largely uncorrelated with the initial weight matrices.
>
> **“The $\eta$ in the SLO update should be a $-\eta$?”**
>
> Yes, thank you for catching this.
>
> **“The motivation for SLO is unclear, as is the statement: "We expect this rule to increase the strength of the UC-NFC in layers where $\ell$ is large relative to $m$ for $m \neq \ell$". Also the appropriate baseline should be to tune all layerwise learning rates, since each layer has a speed. Is there a reason to believe that against this baseline SLO should be better? Is there a way to appropriately set the layerwise speeds?”**
>
> The motivation for SLO is as follows. The uncentered NFC is driven by the C-NFC, hence we expect increasing the magnitude of the centered quantities relative to the initialization should increase the UC-NFC. Further, by forcing the weights in a particular layer to have a higher learning speed than others, we can cause the updates to that weight matrix to be large, while preventing the network from decreasing the loss and the instability in training. We clarify this in the revised manuscript.
>
> SLO can be viewed as adaptively setting learning rates per layer at every epoch. In particular, the learning rate at each layer at each epoch is set to the layerwise learning rate as $\eta_\ell = \frac{C^\ell}{\|\nabla_{W^{(\ell)}} L\|}$. We expect SLO to outperform fixed layerwise learning rates $\eta_\ell=C^\ell$, for example, in the terminal stage of training. At this stage, the loss is small, hence the gradients will be small across all layers, even after scaling by large, fixed learning rates. Instead, SLO will fix the learning speeds independent of the loss scale, allowing for weights to move significantly throughout all of training.

---

### Author Response · Authors · 2024-09-18
**Author response**

We thank the reviewers for their detailed feedback. We have uploaded a revised manuscript with our changes in red text.

In a new Section 4 of our revised manuscript, we prove the correlation will be 1 between the centered NFM and AGOP in a well-studied high dimensional setting - uniformly distributed inputs and activation functions whose derivatives have average value 0 for Gaussian inputs. We show that in this limit they have perfect correlation at initialization. This addresses some reviewer comments by proving our result with detailed conditions, in a way related to previous work on understanding learning in high dimensions.

We have merged the early time analysis from Section 3.1 of our original submission with our new result and our theoretical predictions to form the new Section 4 titled “Theoretical analysis of the C-NFC at early times”.

---

### Decision · Action_Editor_ayFs · 2024-10-29

**Recommendation:** Accept as is

**Comment:**

The paper studies how gradient dynamics may induce feature learning, by analyzing the alignment between the weight gram matrix after training and the AGOP, known as the neural feature ansatz (NFA). This is done through a combination of experiments, theory, and heuristic arguments, which provide some characterization of this phenomenon, particularly near initialization. The reviewers were satisfied with the manuscript and the revisions. The results are interesting and could be useful to the community, therefore I recommend acceptance.

**Audience:**

Yes

**Claims And Evidence:**

Yes